# PTPRN2 and PLCβ1 promote metastatic breast cancer cell migration through PI(4,5)P₂-dependent actin remodeling

Caitlin A Sengelaub, Kristina Navrazhina, Jason B Ross, Nils Halberg & Sohail F Tavazoie[*]

## Abstract

Altered abundance of phosphatidyl inositides (PIs) is a feature of cancer. Various PIs mark the identity of diverse membranes in normal and malignant cells. Phosphatidylinositol 4,5-bisphosphate (PI(4,5)P₂) resides predominantly in the plasma membrane, where it regulates cellular processes by recruiting, activating, or inhibiting proteins at the plasma membrane. We find that PTPRN2 and PLCβ1 enzymatically reduce plasma membrane PI(4,5)P₂ levels in metastatic breast cancer cells through two independent mechanisms. These genes are upregulated in highly metastatic breast cancer cells, and their increased expression associates with human metastatic relapse. Reduction in plasma membrane PI(4,5)P₂ abundance by these enzymes releases the PI(4,5)P₂-binding protein cofilin from its inactive membrane-associated state into the cytoplasm where it mediates actin turnover dynamics, thereby enhancing cellular migration and metastatic capacity. Our findings reveal an enzymatic network that regulates metastatic cell migration through lipid-dependent sequestration of an actin-remodeling factor.

**Keywords** actin; metastasis; phosphoinositides; PI(4,5)P₂; tumor migration
**Subject Categories** Cancer; Cell Adhesion, Polarity & Cytoskeleton; Membrane & Intracellular Transport
The EMBO Journal (2016) 35: 62–76

## Introduction

PIs are major determinants of membrane identity and regulators of membrane trafficking in both normal and disease states (Vicinanza *et al*, 2008; Balla, 2013). The functions of PIs differ greatly depending on the phosphorylation state of the hydroxyl groups on the inositol ring head group, which can be metabolically interconverted to different phosphorylated states by PI kinases and PI phosphatases, or reduced to second messengers through PI hydrolyzers. Through the use of biochemical and cell-biological methods, the cellular roles of various PIs in secretion, endocytosis, actin dynamics, and intracellular signaling have begun to be elucidated. One function of PIs is to serve as docking lipids for the recruitment of specific proteins to cellular compartments, which results in enhancement or inhibition of their activity. PI(4,5)P₂ has previously been shown to regulate cellular migration, a key feature of cancer progression (Clark *et al*, 2000; Sahai & Marshall, 2002; Condeelis & Segall, 2003; Condeelis *et al*, 2005; Ling *et al*, 2006; Luga *et al*, 2012). Enhanced migratory capacity is a required phenotype of metastatic cells, which must move through surrounding tissue, enter the vasculature, and ultimately arrive at and colonize distal organs (Bissell & Radisky, 2001; Chiang & Massague, 2008; Hanahan & Weinberg, 2011). Identifying enzymes that regulate the levels of this lipid in cancer cells could provide additional insights into the mechanisms of cancer cell migration and reveal potential targets for development against metastatic progression. Here, we identify two enzymes, PTPRN2 and PLCβ1, with activity toward PI(4,5)P₂ and that promote cancer cell migration and metastasis in breast cancer.

PTPRN2 was initially identified as an auto-antigen in type I diabetes and is predominantly present in neuroendocrine cells (Lan *et al*, 1996; Lu *et al*, 1996; Wasmeier & Hutton, 1996). As a transmembrane protein, PTPRN2 shuttles between secretory vesicles and the plasma membrane. Due to its presence in neurosecretory vesicles, PTPRN2 has been implicated in insulin and neurotransmitter exocytosis; however, the precise role of PTPRN2 in the secretory pathway is unknown (Cai *et al*, 2011). PTPRN2 belongs to the protein tyrosine phosphatase family, but does not exhibit activity against phosphoprotein substrates due to several critical amino acid variations in the PTP domain (Magistrelli *et al*, 1996). Recently, PTPRN2 was found to exhibit phosphatidylinositol phosphatase (PIP) activity against PI(4,5)P₂ and, to a lesser extent, PI3P (Caromile *et al*, 2010).

PLCβ1 belongs to the family of PLC enzymes, which hydrolyze PI(4,5)P₂ to generate the second messengers diacylglycerol (DAG) and inositol triphosphate (IP3) (Rhee, 2001). PLCβ1 localizes mainly to the inner leaflet of the plasma membrane, where it is activated by the Gaq family of G proteins, although a subset of the protein is found in the cytoplasm and nucleus (Smrcka *et al*, 1991; Taylor *et al*, 1991). Nuclear PLCβ1 has been identified as regulating cellular proliferation and differentiation (Manzoli *et al*, 1997). The

---

Laboratory of Systems Cancer Biology, Rockefeller University, New York, NY, USA
*Corresponding author. Tel: +1 212 327 208; Fax: +1 212 327 7209; E-mail: stavazoie@rockefeller.edu

 

best-characterized member of the PLC family, PLCγ1, has been implicated in oncogenesis through its effects on cell motility and adhesion mediated by IP3 and DAG downstream signaling events (Rebecchi & Pentyala, 2000; Jones *et al*, 2005). We find that PLCβ1, another member of the PLC family, promotes breast cancer migration by reducing plasma membrane PI(4,5)P$_2$.

We identify PTPRN2 and PLCβ1 as enzymes that convergently reduce the abundance of PI(4,5)P$_2$ in the plasma membrane. Through complimentary cell-biological experiments, we find that targeted reduction in plasma membrane PI(4,5)P$_2$ by these enzymes releases plasma membrane-bound cofilin, enhancing actin remodeling in breast cancer cells and increasing their metastatic migration. Our findings reveal novel roles for PTPRN2 and PLCβ1 in cancer cell migration and identify PTPRN2 and PLCβ1 as co-modulators of PI(4,5)P$_2$ in the plasma membrane of cancer cells and as drivers of breast cancer metastasis.

## Results

Given the importance of PI(4,5)P$_2$ in multiple cellular processes, we were intrigued by the finding that transcriptomic profiling of MDA-MB-231 breast cancer cells and their *in vivo*-selected highly metastatic derivative LM2 subline revealed two genes, *PTPRN2* and *PLCB1*, that possess known enzymatic activity for PI(4,5)P$_2$ to be both upregulated at the transcript and protein levels in LM2 cells (Figs 1A and B, and EV1A and B) (Minn *et al*, 2005; Tavazoie *et al*, 2008). We validated the upregulation of these genes in a second independent breast cancer cell line, CN34, and found that both genes exhibited markedly increased expression at the transcript and protein levels in the metastatic CNLM1a derivative subline relative to its parental cell population (Figs 1A and B, and EV1A and B). To functionally test their roles in breast cancer metastasis, we depleted PTPRN2 and PLCβ1 in highly metastatic LM2 cells and performed tail vein metastatic colonization assays. Knockdown of these genes reduced metastatic lung colonization (Fig 1C and D, and Appendix Fig S1A and B). Depletion of either PTPRN2 or PLCβ1 in CnLM1a cells also decreased lung metastatic colonization (Appendix Fig S1C and D). Cells with knockdown of either PLCβ1 or PTPRN2 exhibited significantly reduced signal in the lungs at 24 h post-injection compared to control cells, indicating that knockdown of these genes impacts early stages of metastatic progression (Fig EV1C). To investigate the clinical significance of these genes, we quantified the expression levels of PTPRN2 and PLCβ1 in primary tumor cDNA samples derived from patients diagnosed with various stages of breast cancer. Interestingly, expression levels of both genes increased significantly in tumors of patients with advanced (stage IV) metastatic disease (Fig 1E and F). Furthermore, increased expression of both *PTPRN2* and *PLCB1* associated with significantly worse overall survival (Fig 1G) and worse distal metastasis-free survival (Fig 1H) in two large breast cancer patient cohorts (Gyorffy *et al*, 2010; Cancer Genome Atlas Network, 2012). These findings establish PTPRN2 and PLCβ1 as clinically relevant and functional promoters of breast cancer metastasis.

To identify the mechanism(s) by which PTPRN2 and PLCβ1 mediate metastasis, we investigated several cellular metastatic phenotypes. Depletion of either PTPRN2 or PLCβ1 diminished the ability of cells to invade through Matrigel and to migrate through a

porous trans-well insert (Figs 2A and B, and EV2A and B). However, depletion of neither gene affected cellular proliferation rates, viability, cytotoxicity, or caspase 3/7 activity (Fig EV2C–E and Appendix Fig S2A–C). Cells were further tested for their ability to migrate in a scratch assay. Depletion of either PLCβ1 or PTPRN2 significantly decreased the ability of cells to migrate over a 24-h period (Fig 2C). Interestingly, knockdown of both PTPRN2 and PLCβ1 reduced cellular migratory capacity to a greater extent than knockdown of either single gene (Appendix Fig S2D). Knockdown of PTPRN2 or PLCβ1 in four other breast cancer cell lines (BT-549, CNLM1a, HCC-1806, and MDA-MB-468) also significantly reduced the migratory capacity of these cells (Appendix Fig S2E–L).

To further test the ability of PTPRN2 and PLCβ1 to promote metastasis, we overexpressed these genes in the less metastatic MDA parental breast cancer cells (Fig EV2F and G). Overexpression of either gene was sufficient to increase migration and invasive capacity by at least 50% (Fig 2D–H), without affecting cellular proliferation rates (Fig EV2H). Overexpression of both PTPRN2 and PLCβ1 further increased the migratory capacity of breast cancer cells to a greater extent than overexpression of either gene alone (Appendix Fig S2M). Both PTPRN2 and PLCβ1 are enzymes with previously demonstrated activities for the substrate PI(4,5)P$_2$ (Rebecchi & Pentyala, 2000; Caromile *et al*, 2010). To test whether the enzymatic capacity of these proteins was necessary for the metastatic phenotypes, we mutated the catalytic domain of each protein to generate enzymatically inactive versions. PTPRN2 contains a C(X)$_5$R catalytic domain, common to protein tyrosine phosphatases (Wasmeier & Hutton, 1996; Barford *et al*, 1998). Mutation of the catalytic cysteine residue to the structurally similar serine residue abrogates PTPRN2's ability to dephosphorylate PI(4,5)P$_2$ (Caromile *et al*, 2010); however, this mutation also generates a non-hydrolyzable phospho-serine intermediate in the catalytic domain, trapping the substrate and rendering the enzyme nonfunctional. To dissect the influence of the enzyme's catalytic domain independent of trapping the substrate, we instead mutated PTPRN2's catalytic cysteine residue to an inactive alanine residue to generate PTPRN2[C945A]. We found that while wild-type PTPRN2 overexpression was sufficient to increase invasion, migration, and metastatic lung colonization in mice, these effects were dependent on PTPRN2's catalytic activity as equivalent overexpression of PTPRN2[C945A] failed to enhance these phenotypes (Fig 2D–F).

PLCβ1, as a member of the PLC enzymatic family, contains a highly conserved catalytic domain including a catalytic histidine residue. Mutation of PLCβ1's catalytic H331 residue has been previously shown to abrogate its ability to hydrolyze PI(4,5)P$_2$ (Ramazzotti *et al*, 2008). Overexpression of catalytically inactive PLCβ1[H331Q] failed to increase migration, invasion, and metastasis by breast cancer cells (Fig 2G–I). These findings establish the catalytic activities of PTPRN2 and PLCβ1 as necessary for their pro-metastatic phenotypes.

Given that the enzymatic activities of PTPRN2 and PLCβ1 were required to promote migration and invasion, we next investigated the role of their enzymatic substrate, PI(4,5)P$_2$. PI(4,5)P$_2$ is predominantly present in the plasma membranes of cells, where it has been implicated in various cellular processes (Martin, 2001; Vicinanza *et al*, 2008). Both PTPRN2 and PLCβ1 also demonstrated localization to the plasma membrane in breast cancer cells, in addition to some cytoplasmic localization (Fig EV3A). These enzymes act to

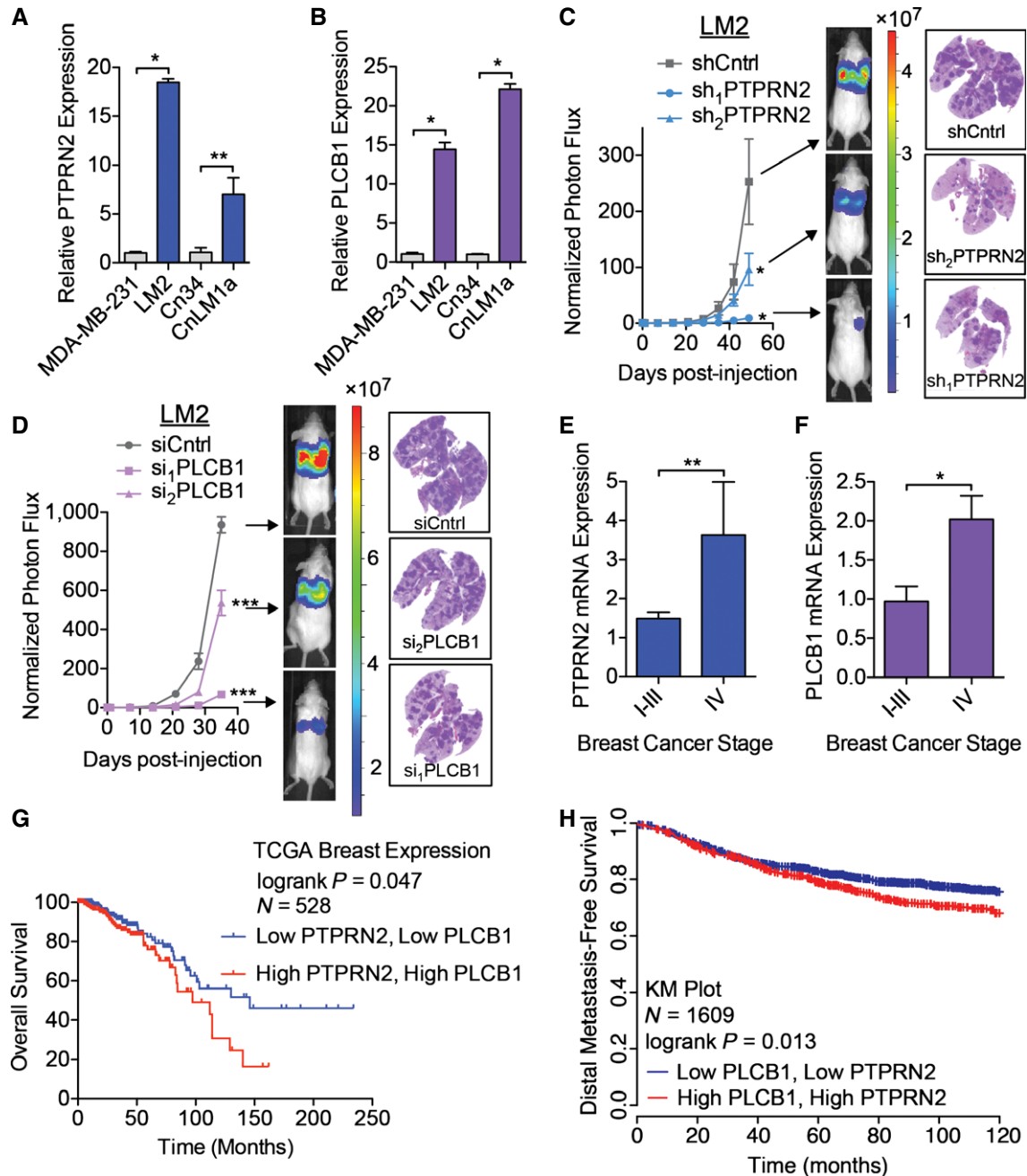

**Figure 1.  PTPRN2 and PLCβ1 promote breast cancer metastasis.**

A, B    *PTPRN2* (A) and *PLCB1* (B) expression levels were determined by qRT–PCR. N = 5.

C    Bioluminescence imaging quantification of lung colonization by 40,000 LM2 breast cancer cells transduced with shRNAs targeting PTPRN2 or a control hairpin. For shCntrl, sh₁PTPRN2: N = 5 mice/group. For sh₂PTPRN2: N = 6 mice. Right, H&E staining of representative lung sections.

D    Bioluminescence imaging quantification of lung colonization by 40,000 LM2 cells transfected with siRNA targeting PLCβ1 or a control siRNA. For siCntrl: N = 5 mice. For si₁PLCβ1, si₂PLCβ1: N = 6 mice/group. Right, H&E staining of representative lung sections.

E, F    *PTPRN2* (E) and *PLCB1* (F) levels were analyzed in human breast cancers (stages I–IV) and normal breast tissue from TissueScan qPCR Array Breast Cancer Panels II and III (Origene, N = 97). Expression levels were normalized to levels in normal tissue for each gene.

G    Kaplan–Meier curve representing overall survival of a cohort of breast cancer patients (N = 528) as a function of their primary tumor's *PTPRN2* and *PLCB1* expression levels (data from the TCGA Research Network, Cancer Genome Atlas Network, 2012). Patients whose primary tumors' *PTPRN2* and *PLCB1* expression levels were higher or lower than the median of the population were classified as low (blue) or high (red) expression.

H    Kaplan–Meier curve representing distal metastasis-free survival of a cohort of breast cancer patients (N = 1,609) as a function of their primary tumor's *PTPRN2* and *PLCB1* expression levels (data from KMPlot, Gyorffy *et al*, 2010). Patients' primary tumors' *PTPRN2* and *PLCB1* expression levels were classified as low (blue) or high (red) expression.

Data information: Error bars represent SEM. *P < 0.05, **P < 0.01, ***P < 0.001.

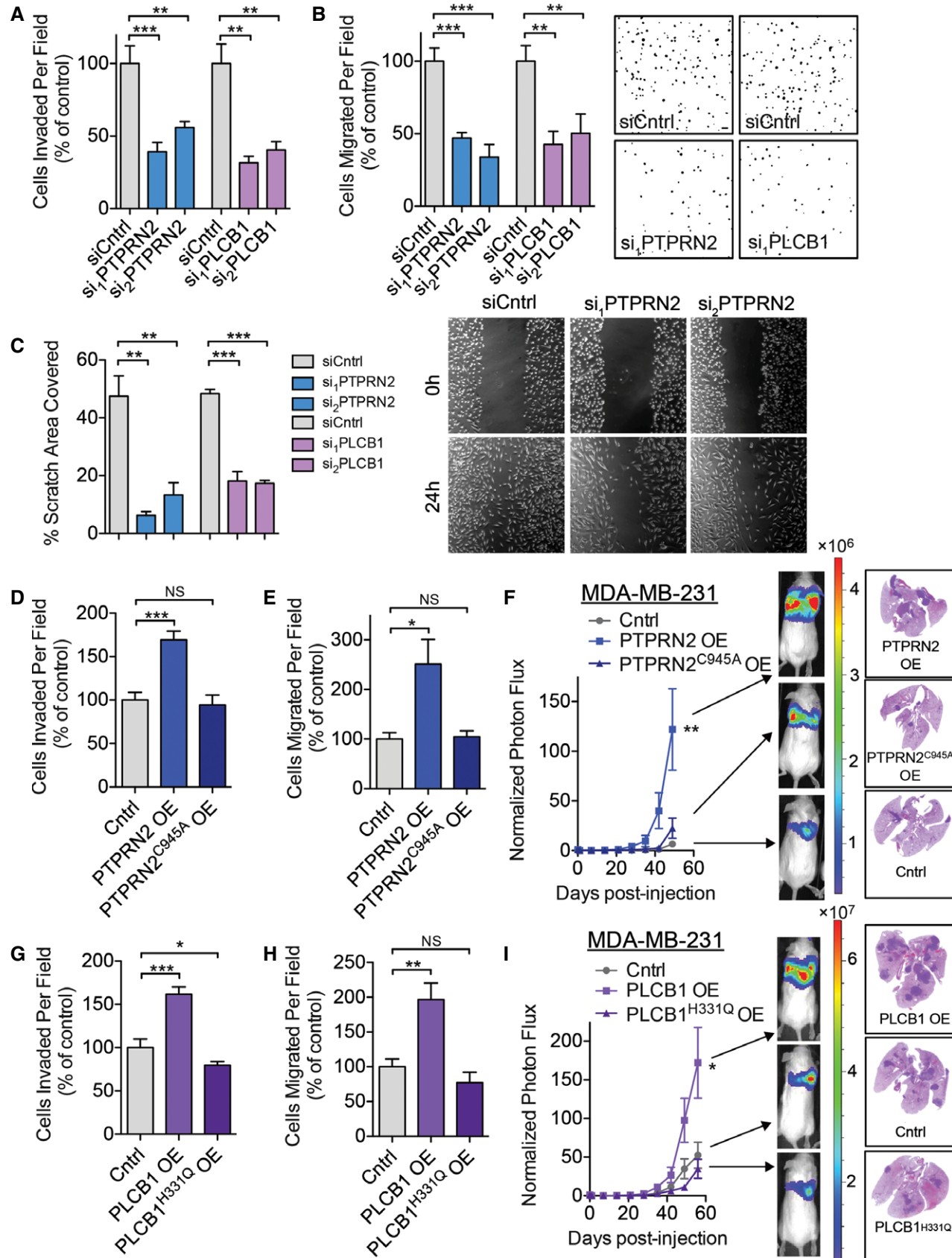

**Figure 2.**

◀

**Figure 2.  PLCβ1 and PTPRN2 drive metastatic migration and invasion.**

A       Matrigel invasion by 50,000 LM2 cells transfected with siRNA targeting PTPRN2, PLCβ1, or control siRNA. Data normalized to control values. *N* = 6 inserts/group.

B       Migration assay by 100,000 LM2 cells transfected with siRNA targeting PTPRN2, PLCβ1, or control siRNA. Data normalized to control values. *N* = 6 inserts/group. Right, representative images of the migration assay. Scale bar, 100 μm.

C       Quantification of area covered by cells 24 h after a scratch was made through confluent cells transfected with siRNA targeting PTPRN2, PLCβ1, or control siRNA. *N* = 5 wells/group. Right, representative images of the scratch assay.

D, E    MDA-MB-231 cells transduced with PTPRN2, PTPRN2$^{C945A}$, or control vector were subjected to the Matrigel invasion (D) and migration assays (E). *N* = 5 inserts/group.

F       Bioluminescence imaging quantification of lung colonization by 40,000 MDA-MB-231 cells overexpressing PTPRN2, PTPRN2$^{C945A}$, or control vector. *N* = 5 mice/group. Right, H&E staining of representative lung sections.

G, H    MDA-MB-231 cells transduced with PLCβ1, PLCβ1$^{H331Q}$, or control vector were subjected to the Matrigel invasion (G) and migration assays (H). *N* = 5 inserts/group.

I       Bioluminescence imaging quantification of lung colonization by 40,000 MDA-MB-231 cells overexpressing PLCβ1, PLCβ1$^{H331Q}$, or control vector. For Cntrl and PLCβ1$^{H331Q}$ OE: *N* = 6 mice/group. For PLCβ1 OE: *N* = 5 mice. Right, H&E staining of representative lung sections.

Data information: Error bars represent SEM. *$P$ < 0.05, **$P$ < 0.01, ***$P$ < 0.001.

reduce the levels of PI(4,5)$P_2$ through two independent mechanisms. PTPRN2 dephosphorylates PI(4,5)$P_2$, while PLCβ1 hydrolyzes PI(4,5)$P_2$ to generate inositol triphosphate (IP3) and diacylglycerol (DAG). We first quantified the abundance of PI(4,5)$P_2$ in the plasma membrane of cancer cells using immunocytochemical techniques previously demonstrated to accurately reflect changes in PI(4,5)$P_2$ mass (Hammond *et al*, 2009, 2012). Interestingly, highly metastatic LM2 cells exhibited lower levels of PI(4,5)$P_2$ in their plasma membranes relative to their less metastatic parental cell population, consistent with increased levels of *PTPRN2* and *PLCB1* in these cells (Fig 3A). We next tested the functional relationship between plasma membrane PI(4,5)$P_2$ levels and metastatic capacity. Addition of exogenous PI(4,5)$P_2$ (Ozaki *et al*, 2000) to LM2 cells prior to intravenous injection reduced the ability of these cells to colonize the lungs of mice relative to cells treated with carrier alone (Figs 3B and EV3B). Addition of exogenous PI(4,5)$P_2$ would be expected to only transiently increase PI(4,5)$P_2$ levels and thus impact early stages of metastatic colonization. Consistent with this, breast cancer cells treated with exogenous PI(4,5)$P_2$ demonstrated reduced metastatic lung signal as early as 24 h post-injection compared to cells treated with carrier alone (Appendix Fig S3A).

Consistent with the activities of PTPRN2 and PLCβ1, overexpression of either PLCβ1 or PTPRN2 reduced membrane levels of PI(4,5)$P_2$ (Figs 3C and EV3C). Conversely, depleting either enzyme in cancer cells increased the membrane levels of PI(4,5)$P_2$ (Fig 3D). We further confirmed our immunofluorescence data by quantifying PI(4,5)$P_2$ levels using an enzyme-linked immunosorbent assay (ELISA). Quantification of PI(4,5)$P_2$ abundance in membrane fractions of cells with overexpression of PTPRN2 or PLCβ1 revealed reduced amounts of the lipid, while depletion of either enzyme increased PI(4,5)$P_2$ quantity in membrane fractions (Appendix Fig S3B). We further tested the impact of PLCβ1 and PTPRN2 modulation on plasma membrane PI(4,5)$P_2$ levels in two additional breast cancer cells lines. Depletion of PTPRN2 or PLCβ1 in MDA-MB-468 and CNLM1a1 breast cancer cells also significantly increased plasma membrane PI(4,5)$P_2$ levels (Appendix Fig S3C and D). These data indicate that PTPRN2 and PLCβ1 regulate plasma membrane levels of PI(4,5)$P_2$ in breast cancer cells.

To test the functional significance of PI(4,5)$P_2$ plasma membrane levels in metastatic migration, we manipulated the levels of this phosphoinositide in cancer cells using two methods. We first sought to determine whether the migration defect of PTPRN2- or PLCβ1-depleted cells could be rescued by decreasing the plasma membrane PI(4,5)$P_2$ levels of these cells. To selectively deplete plasma membrane PI(4,5)$P_2$, we used a rapamycin-induced dimerization system previously developed for this purpose (Heo *et al*, 2006; Varnai *et al*, 2006). In this system, inositol polyphosphate-5-phosphatase E (INPP5E), which depletes PI(4,5)$P_2$ by dephosphorylating it, is fused to FKBP and recruited to the plasma membrane by the constitutively membrane-inserted protein Lyn$_{11}$ fused to FRB. Cancer cells transfected with these constructs and treated with rapamycin showed reduced plasma membrane PI(4,5)$P_2$ staining compared to cells treated with DMSO (Appendix Fig S4A). Adding rapamycin to PTPRN2-depleted or PLCβ1-depleted cells, which displayed greater levels of plasma membrane PI(4,5)$P_2$, significantly enhanced their migratory capacity relative to control cells—effectively rescuing the phenotype (Fig 4A and B).

In a second set of experiments, we increased plasma membrane PI(4,5)$P_2$ levels by adding exogenous PI(4,5)$P_2$. Addition of PI(4,5)$P_2$ using a lipid carrier system increased plasma membrane PI(4,5)$P_2$ levels by 25% (Fig EV3B) (Ozaki *et al*, 2000). Increasing plasma membrane PI(4,5)$P_2$ abrogated the effects of PTPRN2 and PLCβ1 overexpression on migration (Fig 4C and D). These effects were specific since addition of exogenous PI4P, another lipid present in the plasma membrane (D'Angelo *et al*, 2008), had no effect on migratory capacity (Appendix Fig S4B). These data reveal that PTPRN2/PLCβ1-mediated decrease in plasma membrane levels of PI(4,5)$P_2$ correlates with increased metastatic migration capacity.

Given the importance of plasma membrane PI(4,5)$P_2$ abundance on metastatic capacity, we investigated the enzyme upstream of this lipid, PIP5K. PIP5K generates PI(4,5)$P_2$ from PI4P in the plasma membrane (van den Bout & Divecha, 2009). Consistent with our findings that high levels of PI(4,5)$P_2$ decrease the migratory capacity of cancer cells and reduce lung metastasis, breast cancer patients whose tumors expressed high levels of the three isoforms of PIP5K (PIP5K1A, PIP5K1B, and PIP5K1C) experienced increased distal metastasis-free survival compared to patients whose tumors expressed low levels of PIP5K isoforms (Fig 4E) (Gyorffy *et al*, 2010). We focused on the PIP5K1A isoform since this isoform showed reduced expression in highly metastatic LM2 cells compared to poorly metastatic MDA-MB-231 cells (Appendix Fig S4C and D). Overexpression of PIP5K1A in LM2 cells increased plasma membrane PI(4,5)$P_2$ and reduced the ability of these cells to migrate and invade (Fig 4F and G, and Appendix Fig S4E and F). Taken together, these data indicate that the plasma membrane levels of PI(4,5)$P_2$ negatively impact metastatic capacity and that PTPRN2 and PLCβ1 govern the levels of this lipid in breast cancer cells.

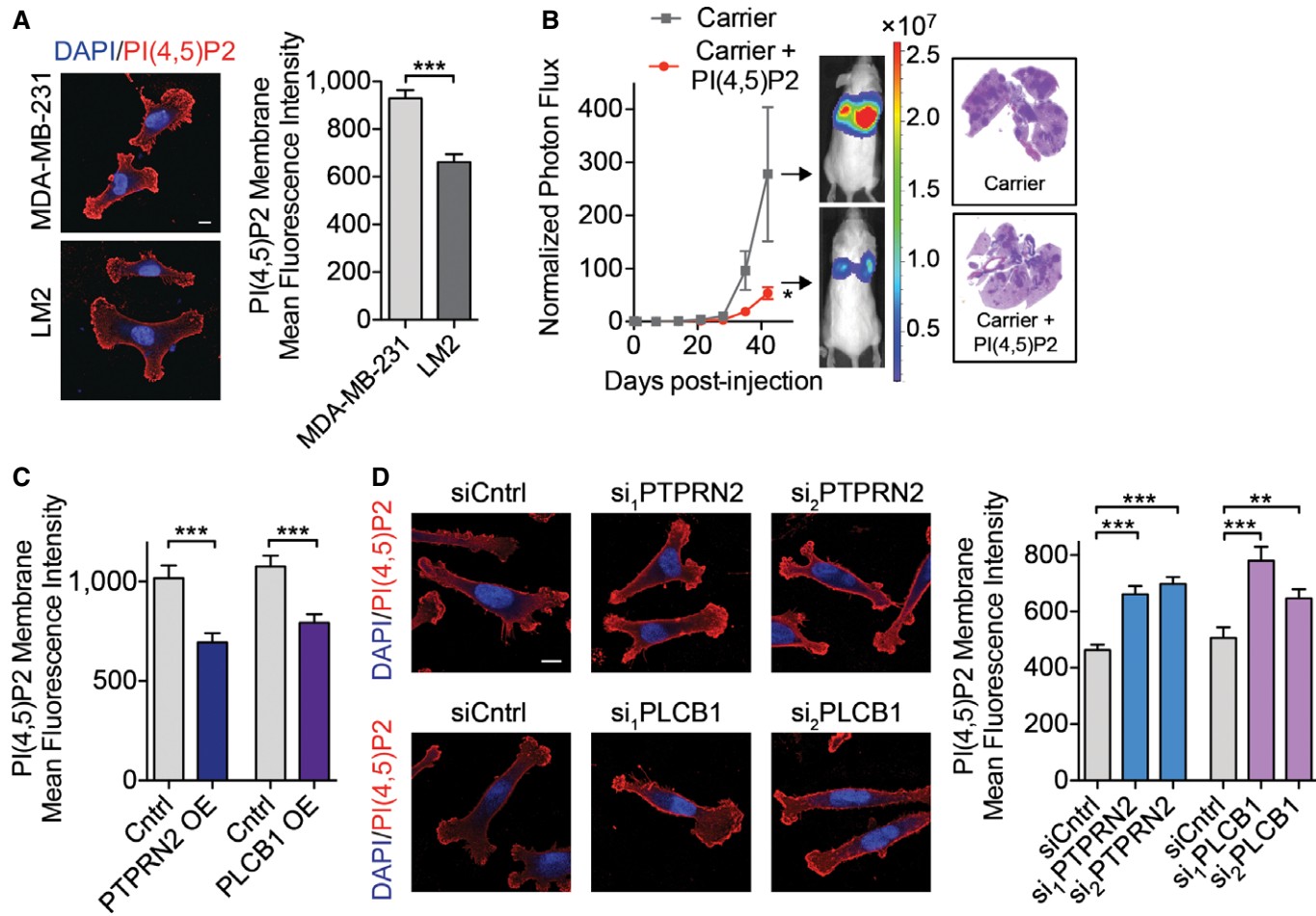

**Figure 3. PLCβ1 and PTPRN2 regulate membrane PI(4,5)P$_2$ levels.**

A  Mean fluorescence intensity of membrane PI(4,5)P$_2$ was analyzed in MDA-MB-231 and LM2 cells immunostained with anti-PI(4,5)P$_2$ antibody using fluorescence microscopy. $N$ = 50 cells/group. Scale bar, 10 μm.

B  Bioluminescence imaging quantification of lung colonization by 40,000 LM2 cells treated with carrier incubated with PI(4,5)P$_2$ or carrier alone for 1 h and then immediately injected. Right, H&E staining of representative lung sections. $N$ = 6 mice/group.

C, D  MDA-MB-231 cells overexpressing PTPRN2, PLCβ1, or a control vector (C) or LM2 cells transfected with siRNA targeting PTPRN2, PLCβ1, or a control siRNA (D) were immunostained for PI(4,5)P$_2$ levels and analyzed by fluorescence microscopy. Mean fluorescence intensity of plasma membrane levels of the lipid was quantified. $N$ = 50 cells/group. Left, representative immunofluorescence images of cells stained with anti-PI(4,5)P$_2$ antibody (red) and 4′,6-diamidino-2-phenylindole (DAPI, blue). Scale bar, 10 μm.

Data information: Error bars represent SEM. **$P$ < 0.01, ***$P$ < 0.001.

Plasma membrane PI(4,5)P$_2$ regulates cellular processes through binding effector proteins, either to recruit these proteins to the plasma membrane or to modulate their activity. Several proteins involved in actin dynamics are known to be inhibited by plasma membrane PI(4,5)P$_2$, including gelsolin, profilin, twinfilin, capping proteins, and cofilin (Saarikangas *et al*, 2010). Of these genes, only the increased expression of cofilin was found to significantly correlate with worse distal metastasis-free survival in a cohort of breast cancer patients (Fig 5A and Appendix Fig S5A–E) (Gyorffy *et al*, 2010). Increased cofilin expression has previously been implicated in breast cancer progression, as well as in oral squamous cellular carcinoma, renal cell carcinoma, and ovarian cancer progression (Martoglio *et al*, 2000; Unwin *et al*, 2003; Wang *et al*, 2004, 2007; Turhani *et al*, 2006). Cofilin binds to plasma membrane PI(4,5)P$_2$, and its membrane binding prevents its ability to bind actin

(Yonezawa *et al*, 1991; Ojala *et al*, 2001; Gorbatyuk *et al*, 2006). When PI(4,5)P$_2$ is hydrolyzed, cofilin is released from the plasma membrane and acts in the cytoplasm as an actin severing protein to promote migration (Ghosh *et al*, 2004; Andrianantoandro & Pollard, 2006; van Rheenen *et al*, 2007). Plasma membrane PI(4,5)P$_2$ levels thus regulate the localization and activation state of cofilin. While we have focused on cofilin as an effector of PTPRN2 and PLCβ1-mediated metastatic migration, we cannot exclude potential roles of other PI(4,5)P$_2$ effector proteins in this pathway. Given that PLCβ1 and PTPRN2 alter the levels of plasma membrane PI(4,5)P$_2$ (Fig 3C and D), we asked whether cofilin localization could also change upon depletion or overexpression of PTPRN2 and PLCβ1. Cells depleted of these enzymes exhibited higher levels of plasma membrane PI(4,5)P$_2$, and thus would be expected to contain increased levels of membrane-associated cofilin. Western blot

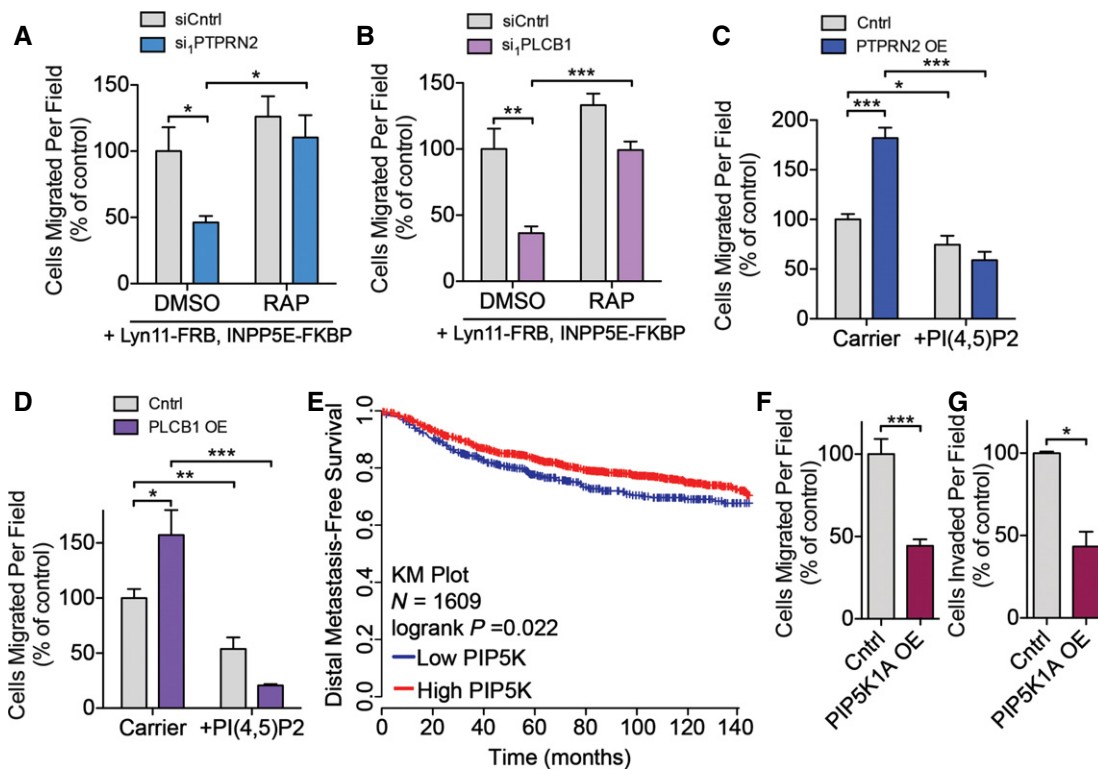

**Figure 4.  Altered PI(4,5)P$_2$ levels modulate metastatic migration and invasive capacity.**

A, B	LM2 cells transfected with siRNA targeting PTPRN2 (A), PLCβ1 (B) or a control siRNA were transfected with Lyn$_{11}$-FRB and INPP5E-FKBP, treated with either DMSO or 100 nM rapamycin and subjected to the migration assay. *N* = 5 inserts/group.

C, D	MDA-MB-231 cells overexpressing PTPRN2 (C), PLCβ1 (D) or control vector were treated with carrier alone or carrier incubated with PI(4,5)P$_2$ for 1 h and then immediately subjected to the migration assay. *N* = 5 inserts/group.

E	Kaplan–Meier curve representing distal metastasis-free survival of a cohort of breast cancer patients (*N* = 1,609) as a function of their primary tumor's mean *PIP5K1A*, *PIP5KB*, and *PIP5KC* expression levels (data from KMPlot, Gyorffy *et al*, 2010). Patients' primary tumors' combined PIP5K expression levels were classified as low (blue) or high (red) expression.

F, G	Migration (F) and Matrigel invasion (G) of LM2 cells transduced with a retroviral vector overexpressing PIP5K1A or control vector. Data normalized to control values. *N* = 5 inserts/group.

Data information: Error bars represent SEM. *P < 0.05, **P < 0.01, ***P < 0.001.

analysis revealed that metastatic cells depleted of either PLCβ1 or PTPRN2 contained significantly more CFL1 in their membrane fractions and less CFL1 in their cytoplasmic fractions compared to control cells (Fig 5B and C, and Appendix Fig S5F–I). Conversely, cells overexpressing these enzymes contained less CFL1 in their membrane fractions and more CFL1 in their cytoplasmic fractions relative to control cells (Fig 5D and E, and Appendix Fig S5J–L). These changes in plasma membrane abundance were independent of changes in whole-cell lysate cofilin abundance, indicating that the amount of CFL1 in the membrane fraction reflected a change in localization rather than expression (Fig EV4A and B). To further confirm the change in cofilin localization upon PTPRN2 or PLCβ1 modulation, we performed immunofluorescence experiments to visualize CFL1 localization. Cells depleted of PTPRN2 or PLCβ1 exhibited significantly increased membrane localization of CFL1 relative to control cells (Fig EV4C), while overexpression of PTPRN2 or PLCβ1 reduced cofilin membrane localization, and increased cytoplasmic staining (Fig EV4D). Taken together, these data indicate that PTPRN2 and PLCβ1 act upstream of CFL1 localization. By decreasing plasma membrane PI(4,5)P$_2$ levels, these

enzymes reduce cofilin's association with the plasma membrane and increase cytoplasmic CFL1 levels.

We next explored whether the CFL1 localization changes observed upon depleting PLCβ1 or PTPRN2 impact CFL1 activity. CFL1 regulates migration by severing F-actin filaments to generate free barbed ends. Generation of free barbed ends by active, non-PI(4,5)P$_2$-bound CFL1 is a necessary step for assembly of actin filaments driving membrane protrusion. Moreover, enhanced abundance of free barbed ends correlates with increased actin polymerization (Chan *et al*, 1998, 2000; Carlsson, 2006). We measured the number of free barbed ends in these cells and observed that cells depleted of PTPRN2 or PLCβ1 contained significantly fewer free barbed ends relative to control cells, as measured by incorporation of labeled actin monomers (Figs 5F and EV4E). These results reveal that PLCβ1 and PTPRN2 regulate breast cancer cell actin polymerization activity—a process dependent on cofilin.

Given the changes in cofilin localization and its activity observed upon PLCβ1 and PTPRN2 modulation, we next asked whether these changes in CFL1 localization alter the actin cytoskeleton in breast cancer cells. Interestingly, cells depleted of PTPRN2 or PLCβ1

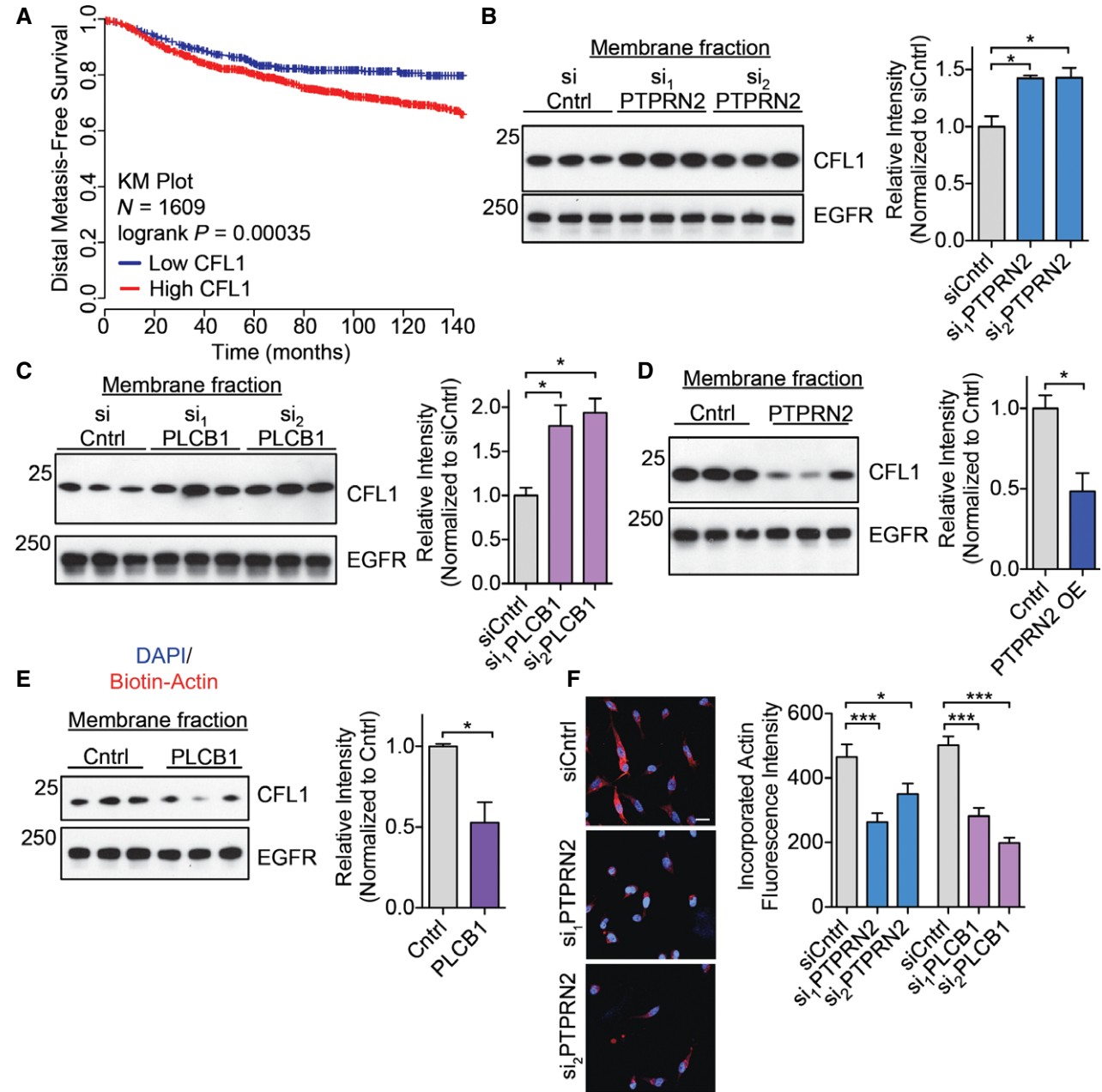

**Figure 5. PLCβ1 and PTPRN2 facilitate cofilin localization and activity.**

A    Kaplan–Meier curve representing distal metastasis-free survival of a cohort of breast cancer patients ($N$ = 1,609) as a function of their primary tumor's *CFL1* expression levels (data from KMPlot, Gyorffy *et al*, 2010). Patients' primary tumors' *CFL1* expression levels were classified as low (blue) or high (red) expression.

B, C    Membrane and membrane-associated proteins were purified from cells transfected with siRNA targeting PTPRN2 (B) or PLCβ1 (C) or control siRNA. Fractions were subjected to Western blot analysis for CFL1 and EGFR levels. Right, densitometry analysis of CFL1 levels normalized to EGFR levels.

D, E    Membrane and membrane-associated proteins were purified from cells overexpressing PTPRN2 (D), PLCβ1 (E) or a control vector. Fractions were subjected to Western blot analysis for CFL1 and EGFR levels. Right, densitometry analysis of CFL1 levels normalized to EGFR levels.

F    LM2 cells transfected with siRNA targeting PTPRN2, PLCβ1, or control siRNA were partially permeabilized and incubated with biotin–actin monomers. Cells were stained for incorporated biotin–actin monomers using Streptavidin-555 (red) and DAPI (blue). Right, quantification of mean fluorescence intensity of incorporated biotin–actin monomers. $N$ = 100 cells/group. Scale bar, 20 μm.

Data information: Error bars represent SEM. *$P$ < 0.05, ***$P$ < 0.001.

exhibited reduced F-actin signal as visualized by immunofluorescence that was independent of total actin abundance (Figs 6A and EV4A). Conversely, PTPRN2 or PLCβ1 overexpression significantly increased cellular F-actin staining intensity (Figs 6B and EV5A). The weak actin filament network seen upon PTPRN2 or PLCβ1 depletion is consistent with the migration defects observed in these cells, since

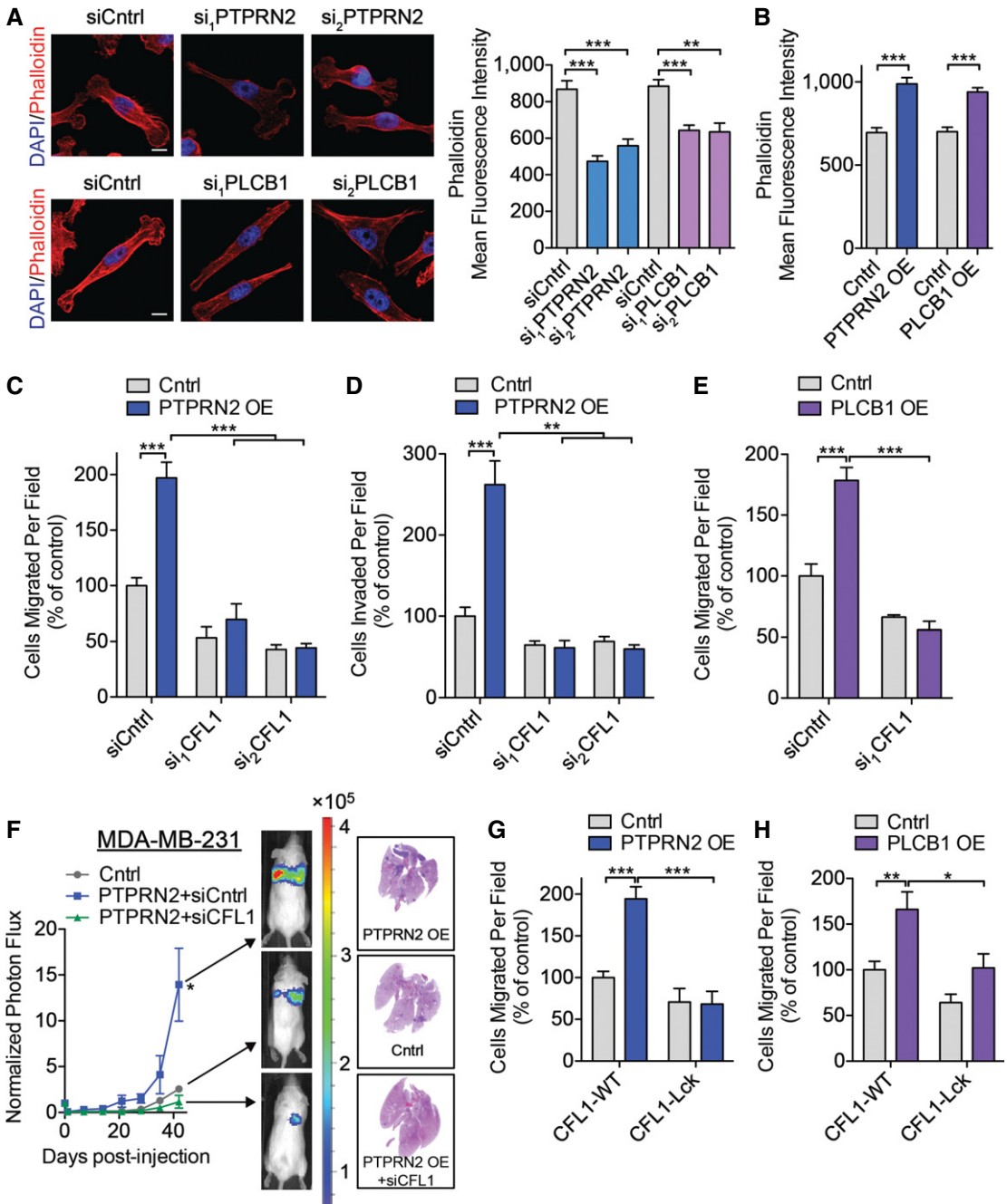

**Figure 6. PLCβ1 and PTPRN2 act upstream of cofilin-mediated actin dynamics.**

A    LM2 cells transfected with siRNA targeting PTPRN2, PLCβ1, or control siRNA were stained with phalloidin (red) and DAPI (blue) and analyzed using fluorescence microscopy. Right, mean fluorescence intensity quantification of whole-cell phalloidin signal. N = 40 cells/group. Scale bar, 10 μm.

B    Mean fluorescence intensity quantification of whole-cell phalloidin signal in MDA-MB-231 cells overexpressing PTPRN2, PLCβ1, or control vector. N = 40 cells/group.

C, D  MDA-MB-231 cells were transfected with siRNAs targeting CFL1 or a control siRNA in the setting of control or PTPRN2 overexpression and subjected to the migration (C) or invasion (D) assays. N = 5 inserts/group.

E    MDA-MB-231 cells were transfected with siRNAs targeting CFL1 or a control siRNA in the setting of control or PLCβ1 overexpression were subjected to the migration assay. N = 5 inserts/group.

F    Bioluminescence imaging quantification of 40,000 MDA-MB-231 cells overexpressing PTPRN2 or control vector and transfected with control siRNA or siRNA targeting CFL1. Right, H&E staining of representative lung sections. N = 5 mice/group.

G, H  MDA-MB-231 cells overexpressing PTPRN2 (G) or PLCβ1 (H) were transfected with siRNA targeting the 3′ UTR of CFL1 to deplete endogenous CFL1. Cells were further transfected with plasmids encoding either GFP-CFL1-WT or GFP-CFL1-Lck and subjected to the migration assay. N = 5 inserts/group.

Data information: Error bars represent SEM. *P < 0.05, **P < 0.01, ***P < 0.001.

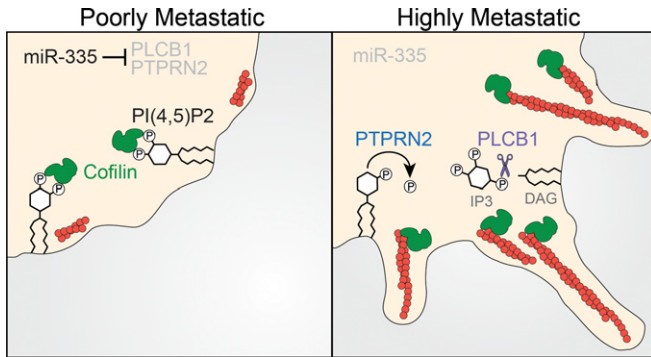

**Figure 7.  PLCβ1 and PTPRN2 promote metastatic breast cancer cell migration as targets of miR-335.**
Model of PTPRN2- and PLCβ1-mediated regulation of plasma membrane PI(4,5)P$_2$ levels to promote metastatic migration through CFL1. In poorly metastatic breast cancer cells, levels of PLCβ1 and PTPRN2 are low. In highly metastatic breast cancer cells, miR-335 is silenced and PTPRN2 and PLCβ1 levels increase. PTPRN2 and PLCβ1 reduce plasma membrane PI(4,5)P$_2$, releasing CFL1 from its inactive state in the plasma membrane to the cytoplasm where it increases actin remodeling.

actin filament polymerization capacity at the leading edge is necessary for cellular migration and invasion (Ghosh *et al*, 2004). These findings are consistent with the reduced F-actin staining intensity and increased membrane-associated (inactive) cofilin in these cells.

The above findings reveal PTPRN2 and PLCβ1 to regulate actin dynamics, decreasing the levels of plasma membrane PI(4,5)P$_2$ and thereby increasing active cytoplasmic cofilin for enhancement of metastatic migration. To establish whether cofilin acts downstream of PLCβ1/PTPRN2-mediated PI(4,5)P$_2$ dynamics and metastatic migration, we performed epistasis experiments. Partial knockdown of CFL1 using RNA-interference abrogated the effects of PTPRN2-mediated enhancement of migration, invasion, and metastatic colonization—consistent with cofilin being necessary for these effects. In the setting of PTPRN2 overexpression, cofilin depletion decreased migration and invasion of breast cancer cells by greater than 70%. Similarly, cofilin depletion reduced migration and invasion capacities of PLCβ1-overexpressing cells (Figs 6C–F and EV5B and C).

We propose that PTPRN2 and PLCβ1 act upstream of cofilin by regulating cofilin's localization through modulation of plasma membrane PI(4,5)P$_2$ levels. Cells depleted of either enzyme exhibit increased plasma membrane PI(4,5)P$_2$ levels and increased membrane-associated cofilin. Given that the activation of cofilin depends on its release from PI(4,5)P$_2$, a constitutively membrane-associated mutant of cofilin should not be able to respond to changes in plasma membrane PI(4,5)P$_2$ levels mediated by PLCβ1 or PTPRN2. We fused the N-terminal sequence of Lck, a Src tyrosine kinase, to cofilin. This myristoylated sequence is sufficient to target proteins to the plasma membrane (Zlatkine *et al*, 1997). Cells transfected with cofilin-Lck exhibited increased cofilin localization to the plasma membrane relative to cells transfected with wild-type cofilin (Fig EV5D). In cells overexpressing either PLCβ1 or PTPRN2, endogenous cofilin was depleted using siRNA targeting the 3′ UTR (Fig EV5B), and cofilin was re-expressed as either wild-type cofilin or membrane-anchored cofilin-Lck. Cells repleted with wild-type cofilin exhibited PTPRN2 and PLCβ1-mediated increases in

migration, while restoring cofilin levels with membrane-anchored cofilin-Lck abolished the ability of PTPRN2 or PLCβ1 to increase migration (Fig 6G and H). These findings indicate that metastatic migration driven by PTPRN2 and PLCβ1 is accomplished through active, non-PI(4,5)P$_2$ membrane-associated CFL1.

*PTPRN2* and *PLCB1* have previously been identified as genes that are negatively regulated by the metastasis suppressor microRNA, miR-335 (Tavazoie *et al*, 2008). MicroRNAs (miRNAs) have been demonstrated to regulate cancer progression through the modulation of metastatic gene networks (Pencheva & Tavazoie, 2013). *PTPRN2* and *PLCB1* expression levels are also clinically correlated with metastatic breast cancer progression (Fig 1E–H). Interestingly, expression levels of *PTPRN2* and *PLCB1* are significantly positively correlated in primary tumors from a cohort of breast cancer patients (Appendix Fig S6A). Western blot analysis revealed decreased PTPRN2 and PLCβ1 protein levels in cells overexpressing miR-335 relative to control cells (Appendix Fig S6B). Our findings support a model wherein the silencing of miR-335 in breast cancer cells enhances expression levels of *PTPRN2* and *PLCB1*. PTPRN2 and PLCβ1 convergently reduce the levels of PI(4,5)P$_2$ in the plasma membrane, dissociating cofilin from the membrane and enabling it to sever cytoplasmic actins to drive actin assembly, metastatic migration, and colonization (Fig 7A).

## Discussion

Tumor cell migration is a key step in successful completion of the metastatic cascade, and PI(4,5)P$_2$ is an important regulator of this process (Condeelis *et al*, 2005). Delineating the mechanisms that govern PI(4,5)P$_2$ levels in cancer cells could provide important insights into the molecular mechanisms that regulate metastatic cell migration. Our findings identify PLCβ1 and PTPRN2 as coregulators of PI(4,5)P$_2$ in the plasma membrane of breast cancer cells. Depletion of PI(4,5)P$_2$ by these proteins increases the metastatic migration of these cells by increasing the abundance of active cytoplasmic cofilin. Increased expression of *PLCB1* and *PTPRN2* correlates with worse overall survival and distal metastasis-free survival in breast cancer patients, further underscoring the clinical relevance of these findings.

The role for PLCβ1 in breast cancer metastasis has not been previously reported; however, *PLCB1* has been identified to be upregulated in colorectal cancer as well (Jia *et al*, 2013). Its family member *PLCG1* has been identified to be overexpressed in metastatic breast cancers (Sala *et al*, 2008). PLCγ1 is activated by the EGF receptor, where it hydrolyzes PI(4,5)P$_2$ to activate cofilin. PLCβ1 differs from PLCγ1 in its activation: PLCβ1 is activated by GPCRs through the G$_q$α subunits of G proteins (Rebecchi & Pentyala, 2000). Our results identify PLCβ1 as an alternative path of cofilin activation in breast cancer—distinct from the route of activation previously ascribed to PLCγ1.

The cofilin/actin depolymerization factor pathway is a well-established mediator of breast cancer invasion and metastasis. Multiple solid cancers exhibit increased expression of cofilin, or altered levels of the kinase LIMK, which phosphorylates cofilin to inhibit its activity (Wang *et al*, 2007). Here, we identify an alternate mechanism utilized by metastatic breast cancer cells to activate cofilin in the absence of EGF stimulation induced PLCγ1

activation, increasing the potential pathways for inducing metastatic migration.

Cofilin is necessary for migration and its increased activity correlates with increased invasiveness in breast cancer cells (Wang *et al*, 2006). Careful temporal and spatial control of cofilin activation is required for optimal formation of protrusions and subsequent migration. Moreover, excessive overexpression of cofilin inhibits actin dynamics (Wang *et al*, 2007). Unregulated PTPRN2 or PLCβ1 activity could deplete $PI(4,5)P_2$ levels and cause loss of membrane polarization and polarized cell signaling. However, there are several mechanisms by which PTPRN2 and PLCβ1 activities are intrinsically restricted. Interestingly, we find that both PTPRN2 and PLCβ1 localize to the plasma membrane, and demonstrate prominent localization to one edge of the plasma membrane, presumably corresponding the leading edge of the cancer cell (Fig EV3A). The restricted localization of these proteins would be predicted to generate spatially restricted cofilin activation, promoting directional migration. Additionally, PTPRN2 cycles between the plasma membrane, vesicles, and the Golgi compartment (Vo *et al*, 2004). Since PTPRN2 is not constitutively located at the plasma membrane, its ability to dephosphorylate plasma membrane $PI(4,5)P_2$ is spatially restricted to one part of its cellular trafficking cycle. PLCβ1 activity is temporally restricted by its requirement for external activation by $G_q\alpha$ proteins. $G_q\alpha$-coupled GPCRs are activated by several small molecules that have been implicated in cancer progression, including CXC chemokines, bradykinin, angiotensin II, and endothelin-1 (Rhee, 2001). The requirement for these small molecules thus restricts the enzymatic activity of PLCβ, and the location of these small molecules in the tumor microenvironment may drive directional cancer cell migration. PLCβ1 activity is further temporally restricted by its intrinsic GTPase stimulating activity, which limits its period of enzymatic activation and prevents continuous depletion of $PI(4,5)P_2$ (Rebecchi & Pentyala, 2000). Together, these mechanisms represent restrictions on the activation of cofilin by PTPRN2 and PLCβ1 to generate optimal plasma membrane protrusion and migration.

PTPRN2 has been previously implicated in the regulated secretion of insulin in pancreatic beta cells and in the regulated secretion of neurotransmitters in neuroendocrine cells, although the precise mechanisms are unknown (Cai *et al*, 2011). Our results do not preclude the possibility of a role for PTPRN2 in constitutive secretion utilized by breast cancer cells, as we observed defects in invasion, a process dependent on secretion of extracellular modifying proteins, upon depletion of PTPRN2 in highly metastatic breast cancer cells. However, our model postulates that the migration-specific defects seen upon PTPRN2 depletion are downstream consequences of cofilin-dependent defects in migration and actin dynamics. The actin cytoskeleton is linked to the *trans*-Golgi, inducing optimal Golgi morphology and facilitating Golgi secretion (Dippold *et al*, 2009). Furthermore, disruption of the actin network using actin-depolymerizing toxins has been shown to inhibit secretion (Lazaro-Dieguez *et al*, 2006).

Here, we have demonstrated the regulation of plasma membrane $PI(4,5)P_2$ by PTPRN2 and PLCβ1, a process that impacts actin dynamics to elicit enhanced metastatic migration. However, $PI(4,5)P_2$ is involved in several other cellular processes. $PI(4,5)P_2$ is the precursor to PI(3,4,5)P3, a key PI in cancer proliferation (Wong *et al*, 2010). $PI(4,5)P_2$ is involved in membrane trafficking through both exocytosis and endocytosis (Martin, 2001; Zoncu *et al*, 2007).

$PI(4,5)P_2$ serves as a docking site for proteins involved in $Ca^{2+}$-triggered vesicle exocytosis; however, depletion of this lipid has also been reported to be necessary for secretion in mast cells (Hammond *et al*, 2006). A similar process has been reported to occur in endocytosis, where $PI(4,5)P_2$ recruits clathrin or other coat proteins, but the lipid must be removed for vesicle fission to complete (Chang-Ileto *et al*, 2011). While our findings focus on the role of PTPRN2 and PLCβ1-mediated depletion of $PI(4,5)P_2$ on actin dynamics and migration, they do not exclude the impact of altered levels of this lipid on additional cellular processes, which may also contribute to metastatic phenotypes through either cancer cell-intrinsic or cancer cell-extrinsic events (Tlsty & Coussens, 2006; Lu *et al*, 2012). Additionally, many other proteins bind $PI(4,5)P_2$ in a manner which either increases or decreases their activity, and our studies do not exclude a role for these proteins as downstream effectors of PTPRN2 and PLCβ1-mediated metastasis. Indeed, it is likely that other cellular pathways are impacted by these changes in $PI(4,5)P_2$ abundance which contribute to metastasis progression. Investigations into the mechanistic roles of $PI(4,5)P_2$ and $PI(4,5)P_2$ effector proteins in these cellular processes during cancer progression will fuel future studies.

## Materials and Methods

### Animal experiments

All animal experiments were conducted in accordance with a protocol approved by the Institutional Animal Care and Use Committee at The Rockefeller University. Mice were housed 5 mice/cage at a 12-h day–night cycle with free access to tap water and food pellets. Six- to 8-week old age-matched female NOD/SCID mice were used for lung metastasis assays. For lung metastatic colonization assays, 40,000 cells in 100 μl PBS were injected via the tail vein as previously described (Png *et al*, 2012).

### Generation of constructs for knockdown and overexpression cell lines

Generation of lentivirus-mediated knockdown and retroviral-mediated overexpression were performed as previously described (Png *et al*, 2012). Primers used to clone overexpression constructs are listed in Appendix Table S1. shRNA sequences are listed in Appendix Table S2. Mutagenesis was performed using QuikChange Lightning Multi Site-Directed Mutagenesis Kit (Agilent Technologies) according to the manufacturer's protocol. Mutagenesis primer sequences are listed in Appendix Table S3.

For $PI(4,5)P_2$ depletion experiments, LYN11-FRB-mcherry (Addgene plasmid # 38004) and PJ-INPP5E (Addgene plasmid # 38001) were gifts from Robin Irvine (Hammond *et al*, 2012).

For cofilin replacement experiments, pEGFP-N1 human cofilin WT was a gift from James Bamburg (Addgene plasmid # 50859) (Garvalov *et al*, 2007).

### Cell culture and transfections

MDA-MB-231, LM2, CN34, and CNLM1a were propagated as previously described (Tavazoie *et al*, 2008; Png *et al*, 2012). BT-549 and

HCC-1806 cell lines (ATCC) were maintained in RPMI medium supplemented with 10% FBS. MDA-MB-468 cells (ATCC) were cultured in DMEM medium containing 10% FBS. All cell lines were routinely tested for mycoplasma contamination.

Transient transfection of siRNAs (Integrated DNA Technologies, sequences in Appendix Table S4) was performed using Lipofectamine 2000 (Life Technologies) according to the manufacturer's protocol. Cells were seeded for invasion and migration assays or subjected to the tail vein metastasis assay 48 h post-transfection. Immunofluorescence was conducted 72 h post-transfection.

Transient transfection of plasmids into cells was conducted using Lipofectamine 2000 according to the manufacturer's protocol. Cells were used for immunofluorescence, migration, or invasion assays 24 h post-transfection.

### RNA extraction and real-time qPCR

Total RNA was collected from cells using Total RNA Purification Kit (Norgen Biotek) according to the manufacturer's protocol. Real-time qPCR was performed as previously described (Tavazoie *et al*, 2008). Primer sequences are listed in Appendix Table S5.

### Proliferation assay

A total of 20,000 LM2 cells were seeded in triplicate for each time point in 6-well plates and counted after 1, 3, and 5 days using a hemocytometer.

### Matrigel invasion assay

Cancer cells were grown to 70% confluence and then the medium was changed to 0.2% FBS DMEM for 16 h. BioCoat Matrigel invasion chambers (Corning) were hydrated with 0.2% DMEM media for 2 h prior to use. Cells were seeded into invasion chambers at 50,000 cells per well in quintuplicate and incubated at 37°C for 20 h. Inserts were rinsed in PBS, and the apical side of the insert was gently scraped to remove non-invaded cells. Inserts were fixed in 4% paraformaldehyde in PBS for 15 min at 37°C. The inserts were excised and mounted with VECTASHIELD HardSet Mounting Medium with DAPI (Vector Laboratories). Invaded cells were imaged using an inverted fluorescence microscope (Zeiss Axiovert 40 CFL). Four images at 10× magnification were taken for each insert and quantified using Fiji software.

### Transwell migration assay

Cancer cells were grown to 70% confluence and medium changed to 0.2% FBS DMEM media for 16 h. Cells were seeded into 3.0 μm PET trans-well migration inserts (Corning) in quintuplicate at 100,000 cells per well in 0.2% FBS DMEM. Inserts were processed and analyzed as described for the Matrigel invasion assay.

### Scratch assay

Cancer cells seeded in triplicate were grown to 90% confluence in 6-well plates and then starved for 16 h in 0.2% FBS DMEM. A scratch through the cell monolayer was made using a 1,000-μl pipette tip. Four images of each well were taken after the scratch was made (0 h) and 24 h later. Images were analyzed using ImageJ (NIH) to quantify the scratch area covered by cancer cells.

### Immunofluorescence

Immunofluorescence detection of $PI(4,5)P_2$ was performed as previously described (Hammond *et al*, 2009). Briefly, cells were fixed in 4% paraformaldehyde, 0.2% glutaraldehyde in PBS for 15 min at room temperature, washed, and then chilled on ice. Cells were blocked and permeabilized for 45 min in 5% goat serum, 50 mM $NH_4Cl$, 0.5% saponin, 20 mM Pipes pH 6.8, 137 mM NaCl, 2.7 mM KCl. Subsequent steps were carried out in this buffer with reduction in saponin to 0.1%. Cells were incubated with anti-$PI(4,5)P_2$ IgM antibody (Echelon Biosciences, Z-B045), followed by Biotinylated anti-IgM antibody (Vector Laboratories), and detected using Streptavidin Alexa Fluor-labeled tertiary antibody (Life Technologies). Cells were post-fixed in 2% paraformaldehyde for 10 min on ice, before warming to room temperature. Cells were rinsed, stained with DAPI, and mounted with ProLong Gold (Life Technologies).

For immunocytochemical detection of CFL1 or FLAG, cells were fixed in 4% paraformaldehyde/PBS for 15 min at room temperature, rinsed, and permeabilized for 5 min with 0.3% Triton-X. Blocking and incubation with antibodies was conducted in 5% goat serum/PBS. Anti-CFL1 antibody (Cell Signaling Technologies, 5175) or anti-FLAG (Sigma, clone M2, F1804) was detected using the appropriate Alexa Fluor-labeled secondary antibody (Life Technologies). Cells were stained with DAPI and mounted in ProLong Gold (Life Technologies).

Immunofluorescence of phalloidin was performed using Alexa Fluor-555 and Alexa Fluor-647 Phalloidin (Life Technologies) according to the manufacturer's protocol.

### Microscopy

Fluorescence images were acquired on an inverted TC5 SP5 laser scanning confocal microscope. Image analysis was performed using Fiji software. To measure mean signal intensity in the membrane compartment, the cell was thresholded and the outline of the cell selected using the magic wand tool. The outline was reduced by 1 μM to form an inner band, and mean signal intensity of this area between the outer and inner band was recorded. For whole-cell phalloidin content, the cell was thresholded and the outline of the cell selected using the magic wand tool. The mean signal intensity of the signal over the area of the cell was recorded.

### Addition of exogenous $PI(4,5)P_2$

Exogenous $PI(4,5)P_2$ was performed using the Shuttle PIP kit (Echelon Biosciences) according to the manufacturer's instructions and as previously described (Ozaki *et al*, 2000). Briefly, $PI(4,5)P_2$-diC16 and Carrier 2 (Histone H1) were solubilized in water to 500 μM concentration. Carrier 2 and $PI(4,5)P_2$ were combined in a 1:1 ratio (100 μM concentration each) and incubated for 15 min at room temperature. The complex was added to cells for 1 h at a final concentration of 10 μM. Treated cells were then subjected to immunofluorescence, migration, or metastatic colonization assays.

## Western blot analysis

Membrane and membrane-associated proteins were isolated from cells using the ProteoExtract native membrane and membrane-associated protein extraction kit (EMD Millipore) according to the manufacturer's instructions. Whole-cell lysate was prepared by lysing cells in RIPA buffer containing protease and phosphatase inhibitors (Roche). Proteins were separated using SDS–PAGE gels (Life Technologies) and transferred to PVDF membrane (EMD Millipore). The following antibodies were used for protein detection: anti-CFL1 (Cell Signaling Technologies, 5175), anti-EGFR (Cell Signaling Technologies, 8504), anti-PTPRN2 (Sigma, HPA006900), anti-PLCβ1 (Sigma, SAB1101340), anti-β-actin (Sigma, A2066), anti-GAPDH (Cell Signaling Technologies, 8884), anti-PIP5K1A (Cell Signaling Technologies, 9693), anti-PIP5K1B (Proteintech, 12541), anti PIP5K1C (Cell Signaling Technologies, 3296). To image loading control proteins, blots were stripped using Restore Western Blot stripping buffer (Thermo Scientific) according to the manufacturer's protocol followed by incubation with appropriate primary antibodies. Bound antibody was detected using the appropriate HRP-conjugated secondary antibodies (Life Technologies). Densitometry analysis of blots was performed using ImageJ (NIH).

## Barbed end assay

Barbed end assay was performed with slight modifications as previously described(Chan *et al*, 1998). Biotin-G-actin (Cytoskeleton, Inc.) was prepared as monomers according to the manufacturer's instructions. Permeabilizing buffer was prepared as 20 mM Hepes, 138 mM KCl, 4 mM $MgCl_2$, 3 mM EGTA, 0.04 g/l saponin, 1 mM ATP, and 1% BSA. Cells were starved for 3 h in 0% FBS DMEM. Cells were treated with addition of serum for 5 min and then incubated in permeabilizing buffer containing 0.2 μM biotin-G-actin for 1 min at 37°C. Cells were rinsed in PBS and fixed for 15 min in 4% paraformaldehyde/PBS. Cells were blocked in 5% goat serum/PBS for 30 min, followed by incubation with Streptavidin-conjugated Alexa Fluor-555 and Phalloidin Alex Fluor-647 (Life Technologies) in blocking buffer for 30 min. Cells were counter-stained with DAPI and mounted using Prolong Gold (Life Technologies). To measure incorporation of biotin–actin monomers in the barbed end assay, the cell was thresholded in the phalloidin signal channel and the outline of the cell selected using the magic wand tool. The mean signal intensity of the streptavidin signal over the area of the cell was recorded.

## Histology

Lungs were prepared by intravenous perfusion of PBS followed by 4% paraformaldehyde/PBS and infusion of paraformaldehyde via the trachea. Removed lungs were embedded in paraffin, sectioned in 5-μm-thick slices, and stained with hematoxylin and eosin.

## Analysis of clinical data sets

Published data generated by the TCGA Research Network (http://cancergenome.nih.gov/) (Cancer Genome Atlas Network, 2012) was used to obtain RNA-Seq expression values for *PTPRN2* and *PLCB1* in breast cancer patients. Values were converted to z-scores

and averaged to determine the *PTPRN2* and *PLCB1* combined gene signature. Each sample was classified as positive for the gene signature if the signal was above the median signal for the population.

KM Plot data from the breast cancer database (version 2014) (Gyorffy *et al*, 2010) was analyzed using JetSet probes only (Li *et al*, 2011). Expression of each gene was calculated using the auto-selected best cutoff. Each sample was classified as positive for the selected gene expression if the signal was above the designated cutoff.

## Statistical analyses and general methods

All statistical analysis was performed using Graphpad Prism 5. For each figure, center bars represent the mean and error bars represent SEM. Populations were determined to be normally distributed by the Kolmogorov–Smirnov normality test, and unpaired Student's one-tailed *t*-test was used to determine significance. *F*-test was performed to compare variances, and Welch's correction was included for populations where the variances differed significantly. For populations where $N < 5$ or populations were not normally distributed, one-tailed Mann–Whitney test was used to determine significance. For Kaplan–Meir survival analysis, a Log-rank (Mantel–Cox) test was used. For correlation of co-expression, Spearman correlation was used. Symbols were used as follows: $*P < 0.05$, $**P < 0.01$, $***P < 0.001$. $P < 0.05$ was considered statistically significant.

For *in vitro* and cellular experiments, no statistical method was used to predetermine sample size. The investigators were not blinded to allocation during experiments and outcome assessment. *In vitro* experiments and imaging experiments were performed a minimum of three independent times with separate culture preparations and imaged in individual sessions. Western blots were conducted three times using independent sample preparations.

For animal experiments, no statistical method was used to predetermine sample size. The investigators were not blinded to allocation during experiments and outcome assessment. Mice were randomized into groups prior to injection. Pre-established criteria for exclusion included accidental death before the completion of the experiment for causes unrelated to the experiment or significant outlier as calculated by sample values greater than two standard deviations from the mean.

**Expanded View** for this article is available online.

## Acknowledgements

We thank members of our laboratory for helpful comments on previous versions of this manuscript. We thank H. Goodarzi for assistance with statistical analysis. We thank C. Alarcon, H. Hang, and S. Simon for insightful discussions. We thank A. North and K. Thomas of The Rockefeller University Bio-Imaging Resource Center for assistance with microscopy. C.A.S. is a recipient of a National Science Foundation Graduate Research Fellowship. K.N. was supported by a Medical Scientist Training Program grant from the National Institute of General Medical Sciences of the National Institutes of Health under award number T32GM007739 to the Weill Cornell/Rockefeller/Sloan-Kettering Tri-Institutional MD-PhD Program. J.B.R. was supported by NIH MSTP grant GM007739. N.H. was supported by a Rockefeller University Anderson Center for Cancer Research Postdoctoral Fellowship. S.F.T. is a Department of Defense Era of Hope Scholar and Breast Cancer Collaborative Scholars and Innovators Award recipient, a Rita Allen Foundation Scholar, and Head of the Elizabeth and Vincent Meyer Laboratory of Systems Cancer Biology.

    

## Author contributions

SFT conceived the project and supervised all research. CAS and SFT wrote the manuscript. CAS and KN designed, performed, and analyzed cell-biological experiments. CAS, JBR and NH designed, performed, and analyzed animal experiments.

## Conflict of interest

The authors declare that they have no conflict of interest.

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
