## [Review Process File · The EMBO Journal]

Manuscript EMBO-2015-91973

PTPRN2 and PLC 1 promote metastatic breast cancer cell migration through PI(4,5)P2-dependent actin remodeling

Caitlin A. Sengelaub, Kristina Navrazhina, Jason B. Ross, Nils Halberg, and Sohail F. Tavazoie

Corresponding author: Sohail Tavazoie, The Rockefeller University

Review timeline:

Submission date:	06 May 2015
Editorial Decision:	10 June 2015
Revision received:	03 September 2015
Editorial Decision:	28 September 2015
Revision received:	16 October 2015
Editorial Decision:	19 October 2015

Editor: Andrea Leibfried

Transaction Report:

1st Editorial Decision

10 June 2015

Thank you for submitting your manuscript entitled 'PTPRN2 and PLC β 1 promote metastatic breast cancer cell migration through PI(4,5)P2-dependent actin remodeling'. I have now received reports from all referees, which are enclosed below.

As you will see, the referees find your study potentially interesting. However, they raise various concerns and do not think that your conclusions are sufficiently supported by the data provided. While referee #1 and #3 offer some support for further consideration of a revised manuscript, referee #2 thinks that your work is unsuitable for publication in The EMBO Journal. Importantly, this referee points out that the observed effects could be due to changes in cell viability upon KD of PTPRN2 and PLCB1/alterations of PIP2 levels and that the proposed regulation of cofilin activity is difficult to reconcile with the existing literature. All referees furthermore note that the proposed long-term effects on PIP2 levels need further and quantitative support.

In light of the partially divergent referee reports, I have asked the referees to comment on each other. Both referee #1 and #2 provided further input (also copied below). Based on this and given the interest into the topic and the constructive comments provided, I can offer to reconsider a revised version, should you be able to address all concerns raised.

Importantly,

- cell viability needs to be monitored to exclude that the observed changes in migration efficiency are due to apoptosis
- better resolved and quantitative data for PIP2 levels need to be provided
- a better mechanistic understanding of how cofilin activity is regulated in your context is needed
- PTPRN2 and PLCB1 protein levels would need to be monitored throughout the experiments under

both overexpression and knock-down conditions

Addressing all concerns raised by the referees clearly demands a lot of work and time, as many of the experiments would have to be repeated and refined, and additional ones would have to be performed as well, with uncertain outcome. So please consider your options carefully and let me know whether you would like to embark into a revision for further consideration at The EMBO Journal. Please also contact me in case of other questions regarding the revision of your manuscript. Thank you for the opportunity to consider your work for publication. I look forward to your revision.

REFEREE REPORTS

Referee #1

This is an interesting and data-rich manuscript by Sengelaub et al reporting the role of two phosphatidylinositol 4,5-bisphosphate (PI4,5P2) metabolizing enzymes (PTPRN2 and PLCB1) in breast cancer cell motility. The enzymes are overexpressed in metastatic breast cancer cell lines and their expression is correlated with poor breast cancer patient outcome. Authors have demonstrated the modulation of PI4,5P2 on plasma membrane by these enzymes promoting the release of cofilin from plasma membrane leading to cofilin-dependent de novo actin polymerization. Authors have incorporated both in vitro and in vivo studies, and appropriate controls are provided supporting the main conclusion of the manuscript. There are some concerns that should be addressed to strengthen the manuscript for publication in EMBO J.

1. Phosphoinositides, lipid signaling molecules are implicated in regulation of different pro-tumorigenic functions, including cell migration. Does the observed phenomenon that decreased PI4,5P2 level by PTPRN2 and PLCB1 and their requirement for cell migration/tumor metastasis occur in other cell types or is this specific for MDA-MB-231 cells?
2. The spatial-temporal generation of phosphoinositide molecules, such as PI4,5P2 and PI3,4,5P3 are generally recognized as the mechanism for facilitating the polarized recruitment of signaling molecules in the leading edges. How do the authors reconcile their findings with literature that have indicated the role for PI4,5P2 and PI3,4,5P3 (product of PI4,5P2) in polarized signaling and cell migration?
3. Authors have illustrated the increased mRNA levels of PTPRN2 and PLCB1 in LM2 and cnLM1a cells, which are the sublines derived from highly metastatic breast cancer cells. Does the expression level of these genes correlates with migratory/invasive capacity of breast cancer cell lines? For example, MCF10A vs. cancer cell lines? Also, the demonstration of up-regulated expression in protein level would be more conclusive.
4. It would interesting to test if there is a correlation of PTPRN2 and PLCB1 with cofilin in a cohort of breast cancer patients (Figure 5).
5. Authors have used MDA-MB-231 cells to demonstrate the migration promoting effects of PTPRN2 and PLCB1. Does overexpression of these genes impart migration promoting effect on non-malignant cells, such as MCF10A?
6. Although the mechanism for PTPRN2 and PLCB1 regulation of cell migration is demonstrated by modulating PI4,5P2 on plasma membrane, does simultaneous knockdown or overexpression of these genes give more profound impact on cell migration or tumor metastasis? Also, is loss of PTPN2 compensated by overexpression of PLCB1 in cell migration and vice versa?
7. It is documented that alteration of single PI4,5P2 metabolizing enzyme has little impact on global PI4,5P2 levels, but for PTPRN2 and PLCB1 this seems not to be the case (Figure 3). It would have been more compelling if authors would have substantiated the IF data with direct PI4,5P2 quantifications. Does overexpression or knockdown of PTPN2 and PLC β 1 bring global changes in PI4,5P2 levels?
8. Overexpression of PIP5K1A reduced cell motility (Figure 4), which is contradictory to the

previous studies (Halsted et al., 2010, J Cell Sci and Chao et al., 2010, J Cell Biol). Is this a cell type specific effect? It may be important to test whether PIP5K1A, B and C protein expression levels are changed in the tested breast cancer cell lines (parental vs. highly metastatic counterpart).

9. A better epistasis experiment would be measuring cell migration after knocking down of PTPRN2 or PLCB1 and testing if overexpression of cofilin rescues the migration defects.

10. The authors' argument in third paragraph of Discussion implies that the PTPRN2/PLCB1/cofilin pathway regulates random migration independently of membrane receptor activation. However, cell migration leading to metastasis is very directional. It would be very interesting to explore the membrane receptor signaling that regulates PTPRN2/PLCB1/cofilin pathway.

11. Model depicted in Figure 7 is poorly described in the figure legends.

Referee #2

In the manuscript of Sengelaub and colleagues, the authors describe how a decrease of PIP2 levels by PTPRN2 and PLCB1 causes cofilin activation with a subsequent actin remodeling and increase in cell migration. Although the subject is of general interest, there are many objections against this study. For example, the authors show that PTPRN2 and PLCB1 overexpression leads to a sustained PIP2 decrease. However, sustained PIP2 depletion leads to apoptosis (e.g. Halstead et al, 2006) which obviously influences their migration and metastasis data. Moreover, it is doubtful whether a sustained decrease of PIP2 levels would increase cofilin activity, since free (non-membrane bound) and non-phosphorylated cofilin is quickly phosphorylated and inactivated. Therefore, sustained PIP2 decrease can never lead to prolonged cofilin activity (hours or even days to weeks as the authors claim). Overall, this manuscript does contain some interesting data, however the mechanisms are too preliminary to meet the high standard required for a high-impact journal such as EMBO journal.

Major comments:

1. The Figures are not in chronological order, which makes the paper difficult to read, and the various figures difficult to compare.

2. Based on Fig 1 and Sup Fig 1, the authors conclude that PTPRN2 and PLCB1 functionally promote metastasis in breast cancer. Although the rest of the paper suggests that this is mediated through inhibition of migration, the effects the authors observe may well be due to a decreased survival rate upon knockdown of the two genes. Moreover, the metastasis assay based on the tail vein injection of tumor cells, is sensitive to cell proliferation/death, but not to migration (the cells are already in the circulation). In addition, there are several issues that need clarification:

- a) Why didn't the authors make stable cell lines? Are cells with a constant knockdown viable?
- b) Does constitutive knockdown affect cell death?
- c) Knockdown of PTPRN2 and PLCB1 are confirmed by qPCR. However, the protein levels are often not correlated with the mRNA levels, especially upon sustained knockdown. The authors need to confirm the decrease of the protein by western blot.
- d) Sup Fig 1 C and D: is the knockdown transient or sustained? It is important to test the RNA and protein levels at the same time scale as sup Fig 2A and B (the same time between the transfection and the 5 days as mentioned in Fig 2A and B).
- e) The authors transfect the cells with siRNA 48 hrs before tail vein injection. Do all cells show a knockdown, or only a fraction of the cells? This data is important to interpret the tail vein injection results.
- f) Fig 2A, B and C: The authors observe a decreased migration potential upon PLCB1 and PTPRN2 knockdown. However, can this data be explained by cells that die upon knockdown?

3. Fig 2 D-H: The authors show that migration is enhanced upon overexpression of PTPRN2 and PLCB1. Based on qPCR the authors conclude that transfection results in a 200 fold overexpression. Can the authors confirm this by western blot?

4. Sup Fig 3A: the authors conclude that PTPRN2 and PLCB1 are predominantly localized at the plasma membrane. This reviewer is not convinced by these images: the proteins are clearly localized in the cytoplasm as indicated by a dark nucleus. To convincingly show that these proteins are

localized at the plasma membrane, the authors should co-transfect e.g. a CFP (that is localized in the cytoplasm), and co-transfect e.g. a plasma membrane tagged GFP protein and show that PTPRN2 and PLCB1 are co-localized with the membrane tagged fluorophore but NOT with the cytoplasm localized fluorophore.

5. PIP2 levels were measured using immunofluorescence detection of PIP2. Microscopy was done on a confocal microscope. The M&M lacks the description of the imaging settings that allows for comparison of various experiments done on different days. For example, how did the authors take in account the fluctuation of lasers (which can be more than 20% over multiple days), and did they keep all other setting equal between experiments (e.g. pinhole, PMT etc). This reviewer would be more convinced when the authors confirm the main PIP2 level results with more standard techniques such as TLC.

6. Fig 3B and Sup Fig 3B: the authors add exogenous PIP2 to cells and show increased potential to metastasize to the lungs. PIP2 levels are well regulated in the cell, and addition of exogenous PIP2 is not likely to be maintained long term by the cell because of dephosphorylation into PIP. Although in Sup Fig 3B the authors show a small increase in PIP2 levels upon exogenous PIP2, it is unclear how long these levels are maintained. What time after addition of exogenous PIP2 are these measurements done? Can the authors illustrate elevated levels of PIP2 at various time points (6, 24, 48hrs, 20 days and 40 days) of metastasis? Without that data, it is difficult to interpret Fig 4C and D.

7. Fig 3D: can the authors illustrate how long the PIP2 levels are altered? Is this a transient effect or a permanent effect?

8. The authors decrease PIP2 levels using a rapamycin-induced recruitment of a phosphoinositide 5-phosphatase enzyme to the plasma membrane. The authors refer to Hammond et al, 2012, but should refer to the original papers (Heo et al., 2006 and Varnai et al., 2006).

9. Fig 4F, G and Sup Fig 4C: can the authors show that PIP2 is increased upon PIP5K1A overexpression?

10. Fig 5 B-E: the authors measure the cofilin abundance in the membrane fraction and see a relationship with the abundance of PIP2. The authors should show that an increase of cofilin in the PM fraction is accompanied by a decrease of cofilin in the cytoplasm and vice versa. Without the internal control, it is difficult to interpret the increase levels of cofilin at the PM.

11. Fig 5F: From the M&M it is not clear how the free barbed ends were measured. What did the authors quantify exactly: total incorporation of biotin-G-actin or the number of filaments with labelled biotin-G-actin?

12. For the cofilin knockdown, the authors should also confirm their knockdown by western blot.

Minor comments:

1) In the main text the authors refer to Sup Fig 5H and Sup Fig 5I which should be 5I and 5J.

Referee #3

PI4,5P2 binding to cofilin prevents actin binding and actin filament severing. Earlier work established that in mammary carcinoma cells, cofilin gets activated by release of a plasma membrane-associated pool upon EGF-induced PI4,5P2 hydrolysis and is required for actin polymerization and cell motility. In this data rich study novel interesting evidence are provided for an alternative pathway based on up-regulation of PLCbeta1 and PTPRN2 leading to increased metastatic potential by lowering PI4,5P2 levels in advanced breast tumors. Experiments are well controlled and results are significant and convincing and this interesting story should be published in the EMBO Journal.

More specifically it would be interesting to check whether increase in lung colonization by exogenously added PI4,5P2 is at the early stage of the metastatic cascade as only short term effect of exogenous PI4,5P2 is expected.

In addition, authors should analyze possible correlation between PLCbeta1 and PTPRN2 levels in

breast cancer as some additive effect of co-up-regulation of the two enzymes on PI4,5P2 level are expected.

Cross-comments:

Referee #1:

The reviewer #2's comments are reasonable and we also raised questions about measuring global PI4,5P2 levels. Although the authors measured PI4,5P2 levels at the plasma membrane, if global PI4,5P2 levels are reduced by knocking down PTPRN2 and PLC β 1 and this leads to cell death as the reviewer #2 pointed out, then the main conclusion of the manuscript would not stand. Thus, it is very important to measure global PI4,5P2 levels and cell death by PTPRN2 and PLC β 1 knockdown as we also noted.

Referee #2:

Although the other two reviewers are more positive, they both seem to have similar concerns as I have. For example, similar to my comments, both referees are concerned about the observed sustained decreases or increases of PIP2 levels, since PIP2 levels are so tightly controlled in cells. Referee#1 mentions that single metabolizing enzymes have generally little impact on global PIP2 levels, and referee#2 expects that exogenous PIP2 can only lead to temporal effects. Especially in the light of metastasis where the authors suggest effects of hours and even days, a sustained change in the PIP2 level are not expected (or cells will just simply die). Referee#1 agrees with me that the IF data should be back up with direct PIP2 quantifications (comment 7).

Lastly, it is important to realize that cofilin activation by releasing a non-phosphorylated pool of cofilin from the membrane by PIP2 decreases only leads to a temporal activation of cofilin. Once the pool of cofilin is released from the membrane, this pool is gone and will be quickly phosphorylated and inactivated. Sustained PIP2 decreases would prevent "reloading" of cofilin to the membrane, so reactivation of cofilin from an empty membrane pool is not possible. Moreover, PIP2 binds to many other proteins including capping proteins, ARPC3 etc, which will all be altered by sustained decreases in PIP2. I suspect that the migration and metastasis effect the authors observe upon PIP2 decreases is simply reflecting the induction of cell death rather than the activation of cofilin.

I would be much more convinced if the authors would:

- 1) Confirm that the overexpression/knockdown is maintained throughout their experiments by western blot (now it is qPCR which does not represent the protein level).
- 2) Confirm their IF data with direct PIP2 quantification (e.g. TLC). It is key to measure the PIP2 levels at various time points of their experiments, to monitor whether the PIP2 changes are really sustained.
- 3) Explain how cofilin activity can be maintained upon sustained PIP2 decreases.

Referee #1:

This is an interesting and data-rich manuscript by Sengelaub et al reporting the role of two phosphatidylinositol 4,5-bisphosphate (PI4,5P2) metabolizing enzymes (PTPRN2 and PLCB1) in breast cancer cell motility. The enzymes are overexpressed in metastatic breast cancer cell lines and their expression is correlated with poor breast cancer patient outcome. Authors have demonstrated the modulation of PI4,5P2 on plasma membrane by these enzymes promoting the release of cofilin from plasma membrane leading to cofilin-dependent de novo actin polymerization. Authors have incorporated both in vitro and in vivo studies, and appropriate controls are provided supporting the main conclusion of the manuscript. There are some concerns that should be addressed to strengthen the manuscript for publication in EMBO J.

Author's Response:

We thank the referee for the constructive and positive comments on our work. We have addressed all the of the referee's comments below and in all cases where experiments could be conducted, have done additional experiments. The referee's suggestions have significantly improved our manuscript.

1. Phosphoinositides, lipid signaling molecules are implicated in regulation of different pro-tumorigenic functions, including cell migration. Does the observed phenomenon that decreased PI4,5P2 level by PTPRN2 and PLCB1 and their requirement for cell migration/tumor metastasis occur in other cell types or is this specific for MDA-MB-231 cells?

Author's Response:

We tested the effect of reduced PTPRN2 or PLC β 1 expression on migration in several additional breast cancer cell lines: CNLM1a1, MDA-MB-468, and BT-549 (Supplemental Figure 2DJ-Q). In each of these cell lines, knockdown of PTPRN2 or PLC β 1 reduced the ability of these breast cancer cells to migrate. To determine if these effects are due to increased PI(4,5)P2, we have quantified the levels of PI(4,5)P2 as modulated by PTPRN2 and PLC β 1 in two additional breast cancer cell lines, CNLM1a1 and MDA-MB-468. Similar to our experiments in MDA-MB-231 cells, we have reduced knocked down PTPRN2 and PLC β 1 levels using siRNA and then quantified the levels of the PI(4,5)P2 levels in the plasma membranes of these cells. We find that reduced PTPRN2 or PLC β 1 expression increases membrane PI(4,5)P2 in both of these additional breast cancer cell lines. We have added this data to Supplementary Figure 3F and G.

2. The spatial-temporal generation of phosphoinositide molecules, such as PI4,5P2 and PI3,4,5P3 are generally recognized as the mechanism for facilitating the polarized recruitment of signaling molecules in the leading edges. How do the authors reconcile their findings with literature that have indicated the role for PI4,5P2 and PI3,4,5P3 (product of PI4,5P2) in polarized signaling and cell migration?

Author's Response:

We thank the referee for this insightful point. Indeed, we recognize the need for spatial restriction of PI(4,5)P2 depletion both for effective actin remodeling and for polarized signaling as discussed by the referee. We discuss several possibilities for restricted localization of enzymatic activity as follows. First, we noted that PTPRN2 and PLC β 1 appear to localize predominantly to the leading edge of the breast cancer cells (Supplementary Figure 3A). PTPRN2 cycles between the Golgi and the plasma membrane, and this recycling would temporally limit its enzymatic activity at the plasma membrane (Vo et al., 2004). PTPRN2 may localize to the leading edge through its transport from the Golgi to the leading edge, similar to the transport of secretory cargo to the leading edge in cancer cells.

Additionally, PLC β 1 requires activation by G proteins, which are activated by G-coupled protein receptors. PLC β 1 is activated downstream of activation of G $_q$ α proteins, which are in turn activated by GPCR-binding of specific chemokines (Rhee, 2001). The requirement of these factors for PLC β 1 activation further spatially restricts the activation and thus enzymatic activity of PLC β 1. Future studies will be required to further determine the exact mechanism by which these enzymatic activities are localized.

Again, we thank the referee for this point and have expanded on this topic in the discussion portion of the manuscript.

The referee raises the additional point that PI(3,4,5)P₃ is a product of PI(4,5)P₂. To address whether PI(3,4,5)P₃ levels or signaling is affected by knockdown or overexpression of PTPRN2 or PLCβ₁, we have additionally measured Akt, an effector protein of PI(3,4,5)P₃, and phospho-Akt levels in cells. Breast cancer cells with knockdown or overexpression of PLCβ₁ or PTPRN2 showed no changes in phospho-Akt (Thr308 or Ser473) levels compared to control cells, suggesting that Akt activation and thus its upstream activator PI(3,4,5)P₃ are not affected by changes in PTPRN2 and PLCβ₁ expression as a consequence of changes in PI(4,5)P₂ levels (figure shown to referees, but removed from review process file). These findings are consistent with previous studies suggesting that different 'pools' of lipids exist within specific cellular compartments and are used for distinct functions. In support of this, membrane PI4P levels do not directly affect membrane PI(4,5)P₂ levels despite the fact that PI4P is a precursor for PI(4,5)P₂ (Hammond et al., 2012, Wuttke et al., 2010).

3. Authors have illustrated the increased mRNA levels of PTPRN2 and PLCB1 in LM2 and cnLM1a cells, which are the sublines derived from highly metastatic breast cancer cells. Does the expression level of these genes correlates with migratory/invasive capacity of breast cancer cell lines? For example, MCF10A vs. cancer cell lines? Also, the demonstration of up-regulated expression in protein level would be more conclusive.

Author's Response:

We thank the referee for this suggestion. We have measured PTPRN2 and PLCβ₁ protein levels in MCF 10A, MDA-MB-231, LM2, CN34, and CNLM1a1 cells. PTPRN2 and PLCβ₁ levels are highest in the highly metastatic LM2 and CNLM1a1 cells relative to their respective poorly metastatic parental cell populations, MDA and CN34. Interestingly, we find the levels of each protein are lowest in the non-tumorigenic MCF 10A cells. We have included this data in Supplementary Figure 1A and B.

To test if increased expression of PTPRN2 or PLCβ₁ correlated with increased migratory capacity, we performed migration assays with MCF 10A, MDA, and LM2 cells (figure shown to referees, but removed from review process file). We found that LM2 cells (with the highest levels of PTPRN2 and PLCβ₁) were the most migratory, MDA cells were less migratory, and MCF 10A cells exhibited the lowest migratory capacity.

4. It would interesting to test if there is a correlation of PTPRN2 and PLCB1 with cofilin in a cohort of breast cancer patients (Figure 5).

Author's Response:

We have analyzed *PTPRN2*, *PLCB1*, and cofilin expression using data from The Cancer Genome Atlas (Cancer Genome Atlas, 2012). We did not find any significant correlation in the expression of *CFL1* with either *PLCB1* or *PTPRN2* (figure shown to referees, but removed from review process file). However this is consistent with our model wherein these proteins change the localization and activation of cofilin, but not its total expression or abundance.

5. Authors have used MDA-MB-231 cells to demonstrate the migration promoting effects of PTPRN2 and PLCB1. Does overexpression of these genes impart migration promoting effect on non-malignant cells, such as MCF10A?

Author's Response:

We have overexpressed PTPRN2 or PLCβ₁ in MCF 10A cells. Interestingly, we find that overexpression of PLCβ₁ increases the migratory capacity of these cells (figure shown to referees, but removed from review process file). Overexpression of PTPRN2 appears to slightly increase migration but these effects were not statistically significant. These results suggest that PLCβ₁ may be capable of promoting migration in MCF 10A cells, but that PTPRN2 may require additional activation signals or proteins found in cancer cells in order for it to promote migration.

6. Although the mechanism for PTPRN2 and PLCB1 regulation of cell migration is demonstrated by modulating PI4,5P2 on plasma membrane, does simultaneous knockdown or overexpression of these genes give more profound impact on cell migration or tumor metastasis? Also, is loss of PTPN2 compensated by overexpression of PLCB1 in cell migration and vice versa?

Author's Response:

We thank the referee for this great suggestion. We have performed all of these experiments suggested by the referee. We find that simultaneous knockdown of PTPRN2 and PLC β 1 reduces migration more than individual knockdown of either single gene. However, it does not completely deplete the migratory capacity of these cells. Simultaneous overexpression of PTPRN2 and PLC β 1 increases migration above than overexpression of either single gene. However, the effects of simultaneous overexpression were not additive or synergistic; migratory capacity was increased by approximately 30% over single-gene overexpression—suggesting an upper-limit on migratory capacity upon modulation of these genes. We have added this additional data to the manuscript in Supplementary Figures 2I and 2U.

We additionally performed the rescue experiments suggested by the referee. We knocked down PTPRN2 and overexpressed PLC β 1, and knocked down PLC β 1 and overexpressed PTPRN2. We found that overexpression of PTPRN2 in the setting of PLC β 1 knockdown rescued the migration capacity to levels that were not statistically significantly different than control cells. Additionally, we found that overexpression of PLC β 1 in the setting of PTPRN2 knockdown partially rescued the migration capacity to approximately 78% the level of control cells (figure shown to referees, but removed from review process file).

7. It is documented that alteration of single PI4,5P2 metabolizing enzyme has little impact on global PI4,5P2 levels, but for PTPRN2 and PLCB1 this seems not to be the case (Figure 3). It would have been more compelling if authors would have substantiated the IF data with direct PI4,5P2 quantifications. Does overexpression or knockdown of PTPN2 and PLC β 1 bring global changes in PI4,5P2 levels?

Author's Response:

We have performed PI(4,5)P2 quantification using an independent method, a PI(4,5)P2 enzyme-linked immunosorbent assay (ELISA) (Echelon Biosciences). We extracted lipids from either whole cells or from isolated membrane fractions of these cells and quantified their levels using a second assay. In the plasma membrane fraction, we find that knockdown of either PLC β 1 or PTPRN2 significantly increases PI(4,5)P2 in this fraction. Conversely, overexpression of PLC β 1 or PTPRN2 reduces membrane fraction PI(4,5)P2 quantity compared to control cells. Quantification of PI(4,5)P2 in whole cells followed similar trends as above but did not reveal significant differences in the overall global quantity of PI(4,5)P2. The differences visualized by immunofluorescence do not reflect global PI(4,5)P2 levels, but only plasma membrane PI(4,5)P2 as the staining protocol utilized only preserves the plasma membrane and does not preserve other cellular membranes. We have added this data in Supplementary Figure 3E.

8. Overexpression of PIP5K1A reduced cell motility (Figure 4), which is contradictory to the previous studies (Halsted et al., 2010, J Cell Sci and Chao et al., 2010, J Cell Biol). Is this a cell type specific effect? It may be important to test whether PIP5K1A, B and C protein expression levels are changed in the tested breast cancer cell lines (parental vs. highly metastatic counterpart).

Author's Response:

We thank the reviewer for this suggestion. In the study by Halstead et al. the authors find that overexpression of PIP5K1A induces neural retraction (Halstead et al., 2010). However, other studies have found that knockdown of PIP5KA induced neurite outgrowth (Liu & Lee, 2013) and that overexpression of PIP5K1A inhibits neurite outgrowth (van Horck et al., 2002, Yamazaki et al., 2002), which is more consistent with our results that overexpression of PIP5K1A reduces migration. The literature on the role of PIP5K isoforms in neurite growth and retraction thus appears to be mixed. The study by Chao et al. uses HeLa cells and fibrosarcoma cells (Chao et al., 2010). However, overexpression of another PIP5K isoform, PIP5KL1, in human gastric cancer cells inhibited their migration (Shi et al., 2010). To determine if the effects we observed may be due to cell type, we performed the experiment suggested by the reviewer, and measured protein levels of PIP5K1A, PIP5K1B, and PIP5K1C in highly metastatic (LM2 and CNLM1a1) and poorly

metastatic (MDA and CN34) breast cancer cells. Interestingly, we found that the levels of PIP5K1A were >60% lower in highly metastatic breast cancer cells compared to poorly metastatic breast cancer cells. The levels of PIP5K1B and PIP5K1C showed slight changes (<20% differences) and were not consistent across both CNLM1a1 and LM2 cells. We have added this data to the manuscript in Supplementary Figure 4C.

9. A better epistasis experiment would be measuring cell migration after knocking down of PTPRN2 or PLCB1 and testing if overexpression of cofilin rescues the migration defects.

Author's Response:

We performed this experiment as suggested by the referee. We transiently expressed CFL1-GFP in cells with knockdown of PTPRN2 or PLC β 1 and control cells and concurrently mock-transfected these cells. All cells were then subjected to the migration assay (figure shown to referees, but removed from review process file). We found that overexpression of cofilin did not rescue the migration defects seen upon knockdown of PTPRN2 or PLC β 1. This is consistent with our model, since additional cofilin added to cells would not necessarily be active cofilin. Indeed, we find that in cancer cells with knockdown of PTPRN2 or PLC β 1, additional cofilin-GFP localizes mainly to the plasma membrane rather than to the cytoplasm (figure shown to referees, but removed from review process file). This is consistent with our model wherein PTPRN2 and PLC β 1 reduce membrane PI(4,5)P2 to allow cofilin to sever actin in the cytoplasm. In the absence of these proteins, additional cofilin alone is not sufficient to rescue the migratory defect phenotype.

10. The authors' argument in third paragraph of Discussion implies that the PTPRN2/PLCB1/cofilin pathway regulates random migration independently of membrane receptor activation. However, cell migration leading to metastasis is very directional. It would be very interesting to explore the membrane receptor signaling that regulates PTPRN2/PLCB1/cofilin pathway.

Author's Response:

As discussed in Point 2, we have expanded our discussion to include possible mechanisms of PTPRN2 and PLC β 1 localization and activation. We believe that further in-depth experimental studies are beyond the scope of the current work and will be the focus of future studies.

11. Model depicted in Figure 7 is poorly described in the figure legends.

Author's Response:

We apologize for poorly described figure legend. We have amended the figure legend for the model figure.

Referee #2:

In the manuscript of Sengelaub and colleagues, the authors describe how a decrease of PIP2 levels by PTPRN2 and PLCB1 causes cofilin activation with a subsequent actin remodeling and increase in cell migration. Although the subject is of general interest, there are many objections against this study. For example, the authors show that PTPRN2 and PLCB1 overexpression leads to a sustained PIP2 decrease. However, sustained PIP2 depletion leads to apoptosis (e.g. Halstead et al, 2006) which obviously influences their migration and metastasis data. Moreover, it is doubtful whether a sustained decrease of PIP2 levels would increase cofilin activity, since free (non-membrane bound) and non-phosphorylated cofilin is quickly phosphorylated and inactivated. Therefore, sustained PIP2 decrease can never lead to prolonged cofilin activity (hours or even days to weeks as the authors claim). Overall, this manuscript does contain some interesting data, however the mechanisms are too preliminary to meet the high standard required for a high-impact journal such as EMBO journal.

Author's Response:

We thank the referee for their careful consideration of our work. We have addressed all of the referee's comments below and have done additional experiments to address their concerns. The referee's suggestions have strengthened our manuscript.

We have addressed each numbered point below. We have addressed the general comments about PI(4,5)P2 and cofilin activity in the Cross Comments section at the end of this document.

Major comments:

1. The Figures are not in chronological order, which makes the paper difficult to read, and the various figures difficult to compare.

Author's Response:

We have ensured that all figures in the manuscript appear in the order they are referred to, and apologize for difficulties these inconsistencies caused.

2. Based on Fig 1 and Sup Fig 1, the authors conclude that PTPRN2 and PLCB1 functionally promote metastasis in breast cancer. Although the rest of the paper suggests that this is mediated through inhibition of migration, the effects the authors observe may well be due to a decreased survival rate upon knockdown of the two genes. Moreover, the metastasis assay based on the tail vein injection of tumor cells, is sensitive to cell proliferation/death, but not to migration (the cells are already in the circulation).

Author's Response:

We agree with the referee that cell death could confound our *in vitro* and *in vivo* results. We previously conducted proliferation assays and found no difference in the proliferation of breast cancer cells with knockdown of PTPRN2 or PLC β 1 (Supplementary Figure 2C, D). We have now further examined the possibility that either transient or sustained knockdown of PTPRN2 or PLC β 1 induces cell death in-depth below (Comment 2f) and found no decreases in cell viability or increases in cell death or apoptosis upon knockdown in these metastatic cells.

We agree with the referee that the cells in the tail vein metastasis assay are already in circulation, however they must possess robust migratory capacity in order to enter the metastatic organ after arresting in the vasculature of the secondary organ. The process of extravasation or trans-endothelial migration at this step is highly dependent upon the migratory capacity of the metastatic cancer cell (Reymond et al., 2013). We agree that the tail vein metastasis assays used in this study do not measure migration of the cancer cell away from the primary tumor, however these assays do reflect the migration of cancer cells from the vasculature into the lungs for metastatic colonization.

In addition, there are several issues that need clarification:

- a) Why didn't the authors make stable cell lines? Are cells with a constant knockdown viable?

Author's Response:

The referee asks why we utilized siRNA-mediated knockdown instead of stable cell lines. In studies with other lipid-modifying proteins, we have found that prolonged knockdown causes other changes in the cell to compensate for the loss of the lipid-modifying protein, and these compensation effects can mask the cell biology regulated by the protein under study. As such, we focused our assays on the use of siRNA, which enables us to study the effects of acute depletion of a protein without the chance for the cell to compensate for loss of the protein.

To address the referee's question if cells with constant knockdown are viable, we constructed shRNA-mediated knockdown cell lines of PTPRN2 and PLC β 1 in LM2 cells and measured the level of shRNA knockdown by western blot (figure shown to referees, but removed from review process file).

To address the issue of their viability, we have conducted proliferation assays, as well as an additional viability, cytotoxicity, and apoptosis assay (discussed in comment 2b below). shRNA-mediated knockdown of either PLC β 1 or PTPRN2 did not affect the proliferation rates of these cells over a 5-day proliferation assay. We have included this data in Supplementary Figure 2E.

b) Does constitutive knockdown affect cell death?

Author's Response:

We have measured cell viability, cytotoxicity, and apoptosis using the ApoTox Glo Triplex assay (Promega). This assay measures cell viability with addition a cell-permeable substrate GF-AFC. GF-AFC is cleaved by a live-cell protease to generate a fluorescent signal proportional to the number of living cells. This assay measures cytotoxicity through the addition of a cell-impermeant substrate bis-AAF-R110. This substrate generates no signal from viable cells, but is cleaved by a dead-cell protease. The viability signal and cytotoxicity signal were unchanged upon shRNA-knockdown of either PLC β 1 or PTPRN2 (figure shown to referees, but removed from review process file). Additionally, we measured caspase 3/7 activity in these cells and again found no significant change in caspase activity upon PTPRN2 or PLC β 1 knockdown (figure shown to referees, but removed from review process file). Together these results indicate that constant knockdown of either gene does not affect cell viability in highly metastatic breast cancer cells.

c) Knockdown of PTPRN2 and PLCB1 are confirmed by qPCR. However, the protein levels are often not correlated with the mRNA levels, especially upon sustained knockdown. The authors need to confirm the decrease of the protein by western blot.

Author's Response:

We thank the referee for this suggestion and agree that western blotting is a better method for confirming knockdown. We have performed western blotting of the shRNA knockdown (see above) and of the siRNA-knockdown (below in response to comment 2d).

d) Sup Fig 1 C and D: is the knockdown transient or sustained? It is important to test the RNA and protein levels at the same time scale as sup Fig 2A and B (the same time between the transfection and the 5 days as mentioned in Fig 2A and B).

Author's Response:

In Supplementary Figure 1C, the knockdown is sustained while in Supplementary Figure 1D the knockdown is transient. Per the referee's suggestion, we have performed western blot analysis of protein levels of either PTPRN2 or PLC β 1 at 1, 3, and 5 days after transient transfection with siRNA targeting the indicated gene. These time points correspond to the time points used in the assay indicated by the referee in Supplementary Figure 2A and 2B. The levels of both genes decrease 1 day post-transfection. The levels of both genes reach their lowest levels 3 days-post transfection (60% and 35% knockdown for PLC β 1, 70% and 75% knockdown for PTPRN2). The levels of PLC β 1 return to their previous levels at day 5. The levels of PTPRN2 remain modestly decreased at day 5. We thank the reviewer for this suggestion, and have added this additional data to Supplementary Figure 2A and B.

e) The authors transfect the cells with siRNA 48 hrs before tail vein injection. Do all cells show a knockdown, or only a fraction of the cells? This data is important to interpret the tail vein injection results.

Author's Response:

We have measured the efficiency of transfection by transfecting highly metastatic breast cancer cells with a small fluorescently labeled oligo (Block-iT Fluorescent Oligo, Life Technologies) using the same quantity and protocol we use to transfect siRNAs. We then performed immunofluorescence and quantified the number of cells successfully transfected with the fluorescent oligo. Greater than 90% of the cells were successfully transfected, indicating the high efficiency of siRNA transfection in highly metastatic breast cancer cells (figure shown to referees, but removed from review process file).

The degree of knockdown will differ on a per cell basis, but is generally accepted to be normally distributed, with some cells showing a very low or very high degree of knockdown while the majority of cells exhibit a degree of knockdown between these two extremes. The western blot data above (Suppl. Fig 2A, 2B) indicates that both PTPRN2 and PLC β 1 were reduced in these cells upon siRNA transfection.

f) Fig 2A, B and C: The authors observe a decreased migration potential upon PLCB1 and PTPRN2 knockdown. However, can this data be explained by cells that die upon knockdown?

Author's Response:

We have previously measured the proliferation capacity of cells with transient knockdown of PTPRN2 or PLC β 1 and found that the number of cells at each time point is not significantly different upon knockdown of either gene compared to control cells (Supplementary Figure 2C, D). We have further tested whether transient knockdown of either gene affects cell death using the ApoTox Glo Triplex assay (Promega). This assay measures cell viability, cytotoxicity, and caspase 3/7 activity. The viability and cytotoxicity signals were unchanged upon siRNA-knockdown of either PLC β 1 or PTPRN2 at 1, 3, and 5 days post-transfection with siRNA. Additionally, we measured caspase 3/7 activity in these cells and again found no change in caspase 3/7 activity upon PTPRN2 or PLC β 1 knockdown by siRNA at 1, 3, and 5 days post-transfection. Together these results indicate that transient knockdown of either gene does not affect cell viability or apoptosis in highly metastatic breast cancer cells. We have added this additional data to Supplementary Figure 2F-H.

To further address the issue of cell viability upon reduced PI(4,5)P₂, we have also tested whether overexpression of either PTPRN2 or PLC β 1 could be promoting metastasis by increasing cellular proliferation or viability, or reducing cell death. We measured 5-day proliferation in breast cancer cells with sustained overexpression of PTPRN2, PLC β 1, or either catalytic mutant and found no differences in the proliferation rates of these cells. We have added this data to Supplementary Figure 2T.

We measured cell viability and cytotoxicity and found no differences in the viability and cytotoxicity among cells with PLC β 1 overexpression, PTPRN2 overexpression, and control cells (figure shown to referees, but removed from review process file). We further measured caspase activity in breast cancer cells with overexpression of either PLC β 1 or PTPRN2 and found no differences in caspase activity between these cells and control cells (figure shown to referees, but removed from review process file).

3. Fig 2 D-H: The authors show that migration is enhanced upon overexpression of PTPRN2 and PLCB1. Based on qPCR the authors conclude that transfection results in a 200 fold overexpression. Can the authors confirm this by western blot?

Author's Response:

We have performed western blot analysis of the overexpression of PTPRN2 and PLC β 1. We find that on the protein level the overexpression of each protein is approximately 2- to 4-fold. We have thus removed the qPCR data from the manuscript and added the western blot data in Supplementary Figure 2R and S, as this provides an accurate indication of the level of overexpression. These levels are much more physiological than the 200-fold overexpression calculated by qPCR, which can be inaccurate, and more consistent with the increased expression in clinical human breast cancer samples as well as the differences we observe in western blot of poorly and highly metastatic breast cancer cells (approximately a 2- to 3-fold increase in highly metastatic cells, Suppl. Fig 1A, B).

4. Sup Fig 3A: the authors conclude that PTPRN2 and PLCB1 are predominantly localized at the plasma membrane. This reviewer is not convinced by these images: the proteins are clearly localized in the cytoplasm as indicated by a dark nucleus. To convincingly show that these proteins are localized at the plasma membrane, the authors should co-transfect e.g. a CFP (that is localized in the cytoplasm), and co-transfect e.g. a plasma membrane tagged GFP protein and show that PTPRN2 and PLCB1 are co-localized with the membrane tagged fluorophore but NOT with the cytoplasm localized fluorophore.

Author's Response:

We agree with the referee that PTPRN2 and PLC β 1 show cytoplasm localization in addition to plasma membrane localization. We do not believe the proteins are localized only to the plasma membrane. We have clarified this point in the manuscript. PTPRN2 has been previously shown to localize to the plasma membrane, Golgi, and secretory vesicles (Vo et al., 2004), so it would be

expected to be found in both the plasma membrane and cytoplasm. PLC β 1 has been previously found to localize to the cytoplasm and plasma membrane as well (Clark et al., 2000). We have performed an additional analysis by staining cells for PTPRN2 or PLC β 1 and EGFR, a receptor known to localize to the plasma membrane. As shown in the images below, we see colocalization of EGFR and PTPRN2 or PLC β 1 in the plasma membrane (figure shown to referees, but removed from review process file).

5. PIP2 levels were measured using immunofluorescence detection of PIP2. Microscopy was done on a confocal microscope. The M&M lacks the description of the imaging settings that allows for comparison of various experiments done on different days. For example, how did the authors take in account the fluctuation of lasers (which can be more than 20% over multiple days), and did they keep all other setting equal between experiments (e.g. pinhole, PMT etc). This reviewer would be more convinced when the authors confirm the main PIP2 level results with more standard techniques such as TLC.

Author's Response:

We agree with the reviewer that lasers may fluctuate. In accordance with the advice from our Bio-Imaging resource facility, we do not compare experiments imaged on separate days. All experimental groups for a given experiment were stained in parallel and then imaged on the same day under identical settings to allow for comparisons between groups. Groups that are directly compared were never imaged on separate days. Experiments were repeated with independent sample preparations, stained on a separate day, and imaged on a different day. Image settings were optimized for a given experimental group to maximize pixel intensity range and prevent signal saturation. Groups from separate experiments cannot be directly compared, although every effort was made to replicate staining and imaging protocols over multiple sessions.

We have performed PI(4,5)P2 quantification using an independent method, a PI(4,5)P2 enzyme-linked immunosorbent assay (ELISA), which has high sensitivity (0.08 pmol). We extracted lipids from membrane fractions of breast cancer cells with either knockdown or overexpression of PLC β 1 or PTPRN2 and quantified PI(4,5)P2 mass. These results were in agreement with the results from our immunofluorescence experiments, wherein depletion of either enzyme increases plasma membrane PI(4,5)P2 abundance while overexpression decreases lipid abundance. We have added this additional data to Supplementary Figure 3E.

6. Fig 3B and Sup Fig 3B: the authors add exogenous PIP2 to cells and show increased potential to metastasize to the lungs. PIP2 levels are well regulated in the cell, and addition of exogenous PIP2 is not likely to be maintained long term by the cell because of dephosphorylation into PIP. Although in Sup Fig 3B the authors show a small increase in PIP2 levels upon exogenous PIP2, it is unclear how long these levels are maintained. What time after addition of exogenous PIP2 are these measurements done? Can the authors illustrate elevated levels of PIP2 at various time points (6, 24, 48hrs, 20 days and 40 days) of metastasis? Without that data, it is difficult to interpret Fig 4C and D.

Author's Response:

We agree with the referee that exogenously added PI(4,5)P2 is likely to only have a short-term effect due to metabolic conversion. The measurements in Supplementary Figure 3B were made after the cells had been incubated with exogenous PI(4,5)P2 for one hour, as this is also the point at which the cells were then injected into mice or used in *in vitro* assays. We have modified the figure legend to clarify this.

Per the referee's suggestion, we have measured PI(4,5)P2 levels at 1, 6, 12, and 24 hours (the 1 hour time point corresponds to directly after the 1 hour incubation period) (figure shown to referees, but removed from review process file). We observe a significant increases in PI(4,5)P2 levels at 1 hour, and a slight increase at 6 and 12 hours (but are not statistically significant).

We support the idea that the addition of exogenous PI(4,5)P2 affects the very early stages of metastatic colonization following tail vein injection. Cancer cells injected via the tail vein immediately arrest in the capillary beds of the lungs, but then must immediately extravasate in order

to enter the lungs. This requires that cancer cells possess the ability to efficiently migrate. Although the effects of exogenous PI(4,5)P2 are transient, additional PI(4,5)P2 would be still be present in the cells during this process, as cancer cells were injected immediately following incubation with exogenous lipid. Indeed, cancer cells in this experiment showed reduced lung colonization at one day post-injection, consistent with an early effect of additional PI(4,5)P2 in metastatic colonization. This decreased signal at day 1 is likely the underlying cause of the decreased signal at the endpoint of the metastasis assay, rather than any long-term effects of exogenous PI(4,5)P2 since these levels are returned to their normal levels by 12 hours post-addition. Thus, if within the first 1-2 hours, only 50% of the cells are able to migrate into the organ and escape microvascular trauma-induced death (Furlow et al., 2015), a significant reduction in metastatic signal at the end-point of the experiment will be observed. We thank for referee for addressing this point, and have clarified this point in the manuscript and added the additional data to Supplementary Figure 3C.

7. Fig 3D: can the authors illustrate how long the PIP2 levels are altered? Is this a transient effect or a permanent effect?

Author's Response:

The levels of PI(4,5)P2 in Figure 3D were measured in cells with transient knockdown of PLC β 1 or PTPRN2 at 3 days post-transfection. To determine if the increase in PI(4,5)P2 levels is a transient effect, we have analyzed levels of PI(4,5)P2 in the cells at 1 day and 5 days post-transfection with siRNA targeting PLC β 1 or PTPRN2. Consistent with the decrease in PTPRN2 or PLC β 1 protein at 1 day post-transfection, cells with knockdown of either gene showed slightly increased PI(4,5)P2 levels at this time point (figure shown to referees, but removed from review process file). At 5 days post-transfection, there was no difference in PI(4,5)P2 levels (figure shown to referees, but removed from review process file). This is consistent with the alteration in PI(4,5)P2 levels being a transient effect.

8. The authors decrease PIP2 levels using a rapamycin-induced recruitment of a phosphoinositide 5-phosphatase enzyme to the plasma membrane. The authors refer to Hammond et al, 2012, but should refer to the original papers (Heo et al., 2006 and Varnai et al., 2006).

Author's Response:

We apologize for the error and have amended the text to reflect the correct references.

9. Fig 4F, G and Sup Fig 4C: can the authors show that PIP2 is increased upon PIP5K1A overexpression?

Author's Response:

We thank the referee for their suggestion of this important control. We have analyzed membrane PI(4,5)P2 levels in cells with PIP5K1A overexpression, and find that these levels are significantly increased compared to control cells. We have added this data to Supplementary Figure 4F.

Additionally we have performed a western blot to quantify the level of PIP5K1A overexpression and added this data the Supplementary Figure 4E.

10. Fig 5 B-E: the authors measure the cofilin abundance in the membrane fraction and see a relationship with the abundance of PIP2. The authors should show that an increase of cofilin in the PM fraction is accompanied by a decrease of cofilin in the cytoplasm and vice versa. Without the internal control, it is difficult to interpret the increase levels of cofilin at the PM.

Author's Response:

We have performed western blot analysis of cofilin levels in the cytoplasm fraction of cells with knockdown or overexpression of PLC β 1 or PTPRN2. Consistent with our results in the membrane fractions of these cells, we find that upon knockdown of PLC β 1 or PTPRN2 cytoplasmic cofilin decreases in breast cancer cells. Conversely, overexpression of PLC β 1 or PTPRN2 significantly

increases cofilin in the cytoplasmic fraction of these cells. We have added this data to Supplementary Figures 5G-L.

11. Fig 5F: From the M&M it is not clear how the free barbed ends were measured. What did the authors quantify exactly: total incorporation of biotin-G-actin or the number of filaments with labelled biotin-G-actin?

Author's Response:

We apologize for the lack of detailed description in the methods section regarding this method. We have measured total incorporation of biotin-G actin by measuring the fluorescent signal from Streptavidin Alex-Fluor conjugate. We have amended the Methods section to clarify this.

12. For the cofilin knockdown, the authors should also confirm their knockdown by western blot.

Author's Response:

We have performed western blot analysis of cofilin knockdown. We find each siRNA reduces cofilin protein levels by greater than 60%. We have added this additional data to the manuscript in Supplementary Figure 6B.

Minor comments:

- 1) In the main text the authors refer to Sup Fig 5H and Sup Fig 5I which should be 5I and 5J.

Author's Response:

We regret the error and have corrected the numbering in the text. We thank the referee for all of their helpful suggestions.

Referee #3:

PI4,5P2 binding to cofilin prevents actin binding and actin filament severing. Earlier work established that in mammary carcinoma cells, cofilin gets activated by release of a plasma membrane-associated pool upon EGF-induced PI4,5P2 hydrolysis and is required for actin polymerization and cell motility. In this data rich study novel interesting evidence are provided for an alternative pathway based on up-regulation of PLCbeta1 and PTPRN2 leading to increased metastatic potential by lowering PI4,5P2 levels in advanced breast tumors. Experiments are well controlled and results are significant and convincing and this interesting story should be published in the EMBO Journal.

More specifically it would be interesting to check whether increase in lung colonization by exogenously added PI4,5P2 is at the early stage of the metastatic cascade as only short term effect of exogenous PI4,5P2 is expected.

Author's Response:

We thank the referee for their kind comments on our efforts to delineate the mechanism of PTPRN2 and PLCβ1 in breast cancer metastasis.

Per the referee's suggestion, we have examined lung colonization at one day post-injection of cancer cells treated with either exogenous PI(4,5)P2 or control cells (figure shown to referees, but removed from review process file). We find that even at this early time point, cells treated with exogenous PI(4,5)P2 show significantly reduced lung colonization compared to control cells. Indeed, given that exogenously added PI(4,5)P2 would only be expected to have a short-term effect on the metastatic capacity of cells, we have added this data to the manuscript in Supplementary Figure 3C. This effect at early time points is likely the underlying reason for the difference in metastatic colonization at the endpoint of the assay (day 42 post-injection) rather than any long-term effects of exogenous PI(4,5)P2. We thank the reviewer for bringing up this point, and we have clarified this point in the manuscript.

In addition, authors should analyze possible correlation between PLCbeta1 and PTPRN2 levels in breast cancer as some additive effect of co-up-regulation of the two enzymes on PI4,5P2 level are expected.

We have analyzed patient data from The Cancer Genome Atlas (Cancer Genome Atlas, 2012) and interestingly find that the expression of these genes in primary tumors from breast cancer patients are significantly positively correlated. We have added this data to the manuscript in Supplementary Figure 7A.

Cross-comments:

Referee #1:

The reviewer #2's comments are reasonable and we also raised questions about measuring global PI4,5P2 levels. Although the authors measured PI4,5P2 levels at the plasma membrane, if global PI4,5P2 levels are reduced by knocking down PTPRN2 and PLC β 1 and this leads to cell death as the reviewer #2 pointed out, then the main conclusion of the manuscript would not stand. Thus, it is very important to measure global PI4,5P2 levels and cell death by PTPRN2 and PLC β 1 knockdown as we also noted.

Author's Response:

We have performed these experiments as suggested by reviewer 1 and 2. We have measured proliferation, cell viability, cytotoxicity, and apoptosis markers in cells with both transient and sustained knockdown of PLC β 1 or PTPRN2 and observed no differences between these cells and control cells. We have furthermore measured proliferation, cell viability, cytotoxicity, and apoptosis markers in cells with overexpression of PLC β 1 or PTPRN2 and find no changes between these cells and control cells. These data are inconsistent with the effects we observe in migration and metastasis being due to differences in cell viability.

As discussed above, we have measured PI(4,5)P2 levels using a second assay, a PI(4,5)P2 ELISA. We find differences in PI(4,5)P2 abundance in the membrane fractions of cells consistent with our immunofluorescence data.. We find modest but not statistically significant differences in global PI(4,5)P2 quantity..

Referee #2:

Although the other two reviewers are more positive, they both seem to have similar concerns as I have. For example, similar to my comments, both referees are concerned about the observed sustained decreases or increases of PIP2 levels, since PIP2 levels are so tightly controlled in cells. Referee#1 mentions that single metabolizing enzymes have generally little impact on global PIP2 levels, and referee#2 expects that exogenous PIP2 can only lead to temporal effects. Especially in the light of metastasis where the authors suggest effects of hours and even days, a sustained change in the PIP2 level are not expected (or cells will just simply die). Referee#1 agrees with me that the IF data should be back up with direct PIP2 quantifications (comment 7).

Lastly, it is important to realize that cofilin activation by releasing a non-phosphorylated pool of cofilin from the membrane by PIP2 decreases only leads to a temporal activation of cofilin. Once the pool of cofilin is released from the membrane, this pool is gone and will be quickly phosphorylated and inactivated. Sustained PIP2 decreases would prevent "reloading" of cofilin to the membrane, so reactivation of cofilin from an empty membrane pool is not possible. Moreover, PIP2 binds to many other proteins including capping proteins, ARPC3 etc, which will all be altered by sustained decreases in PIP2. I suspect that the migration and metastasis effect the authors observe upon PIP2 decreases is simply reflecting the induction of cell death rather than the activation of cofilin. I would be much more convinced if the authors would:

- 1) Confirm that the overexpression/knockdown is maintained throughout their experiments by western blot (now it is qPCR which does not represent the protein level).

Author's Response:

We have performed western blot data to confirm our knockdown and overexpression and have added these to the manuscript.

- 2) Confirm their IF data with direct PIP2 quantification (e.g. TLC). It is key to measure the PIP2 levels at various time points of their experiments, to monitor whether the PIP2 changes are really sustained.

Author's Response:

As discussed above, we have measured PI(4,5)P2 levels using a second assay, a PI(4,5)P2 ELISA in the membrane fractions of cells. These data were in agreement with our immunofluorescence results. Additionally, we have measured PI(4,5)P2 levels over the course of our experiments and found that these changes are sustained for 1-3 days upon transient knockdown of either PTPRN2 or PLC β 1. After this point, the levels return to the control levels. Thus PI(4,5)P2 changes are sustained throughout the course of our *in vitro* experiments (1-2 days) and the early points of the metastasis assay (injection at 48 hours post-transfection and 1 day post-injection corresponding to 72 hours post-transfection). These findings are consistent with our hypothesis that PI(4,5)P2 levels affect the migration necessary at the very early stages in this assay such that cancer cells are able to extravasate out of the vasculature and into the lung.

3) Explain how cofilin activity can be maintained upon sustained PIP2 decreases.

Author's Response:

We agree with the referee sustained and complete depletion of PI(4,5)P2 would not allow for sustainable cofilin activity and would impact other proteins that bind PI(4,5)P2. However we do not claim that PLC β 1 or PTPRN2 constitutively decrease PI(4,5)P2, nor do our results show that they remove the vast majority of PI(4,5)P2 from the plasma membrane. We instead support the notion that PTPRN2 and PLC β 1 activity is likely to be spatially and temporally restricted to allow for optimal localized cofilin activation. Indeed, PTPRN2 and PLC β 1 activity is regulated through several mechanisms. PTPRN2 is transported to the plasma membrane and then recycled back to the Golgi in a cyclical manner (Vo et al., 2004). Since PTPRN2 is not constitutively localized to the plasma membrane, it is only able to dephosphorylate PI(4,5)P2 during one part of its cellular trafficking cycle, restricting its activity. PTPRN2 may be localized to the leading edge of the cancer cells by similar mechanisms which direct secretory cargo, such as metalloproteinases, to the leading edge of cancer cells for secretion.

PLC β 1 activity is temporally restricted by its requirement for external activation. The PLC β family is activated by heterotrimeric G proteins, and PLC β 1 is specifically activated by the G $_q\alpha$ subunits. G $_q\alpha$ -coupled GPCRs are activated by several molecules that have been implicated in cancer, including CXC chemokines (implicated in metastatic angiogenesis), bradykinin (identified as a growth factor in multiple cancer types), angiotensin II (promotes migration and proliferation in breast and lung cancer), and endothelin-1 (overexpressed in multiple cancer types, promotes survival and migration) (Rhee, 2001). The fact that these molecules must be present to activate PLC β 1 restricts its enzymatic activity. The localization of these activating molecules in the microenvironment may perhaps also provide a chemotactic directional effect on cellular migration. PLC β 1 activity is further temporally restricted by the length of G protein activation. Activated G proteins contain intrinsic GTPase activity, quickly inactivating the signaling cascade. Interestingly, PLC β enzymes possess GTPase stimulating activity, or the ability to increase the rate of inactivation of the G protein to prevent constitutive activation of enzymatic activity (Rebecchi & Pentylala, 2000).

These mechanisms prevent constitutive depletion of cellular PI(4,5)P2 by either PLC β 1 or PTPRN2, allowing for cofilin to bind membrane PI(4,5)P2. The controlled activity of these proteins is indeed highly important for their role in migration.

We apologize that we did not better clarify this point in the previous version of the manuscript, and have expanded the discussion section of the manuscript to include these points.

REFERENCES

- Cancer Genome Atlas N (2012) Comprehensive molecular portraits of human breast tumours. *Nature* 490: 61-70
- Chao WT, Daquinag AC, Ashcroft F, Kunz J (2010) Type I PIPK-alpha regulates directed cell migration by modulating Rac1 plasma membrane targeting and activation. *J Cell Biol* 190: 247-62
- Clark EA, Golub TR, Lander ES, Hynes RO (2000) Genomic analysis of metastasis reveals an essential role for RhoC. *Nature* 406: 532-5
- Furlow PW, Zhang S, Soong TD, Halberg N, Goodarzi H, Mangrum C, Wu YG, Elemento O, Tavazoie SF (2015) Mechanosensitive pannexin-1 channels mediate microvascular metastatic cell survival. *Nat Cell Biol* 17: 943-52

- Halstead JR, Savaskan NE, van den Bout I, Van Horck F, Hajdo-Milasinovic A, Snell M, Keune WJ, Ten Klooster JP, Hordijk PL, Divecha N (2010) Rac controls PIP5K localisation and PtdIns(4,5)P(2) synthesis, which modulates vinculin localisation and neurite dynamics. *J Cell Sci* 123: 3535-46
- Hammond GR, Fischer MJ, Anderson KE, Holdich J, Koteci A, Balla T, Irvine RF (2012) PI4P and PI(4,5)P2 are essential but independent lipid determinants of membrane identity. *Science* 337: 727-30
- Liu T, Lee SY (2013) Phosphatidylinositol 4-phosphate 5-kinase alpha negatively regulates nerve growth factor-induced neurite outgrowth in PC12 cells. *Exp Mol Med* 45: e16
- Rebecchi MJ, Pentylala SN (2000) Structure, function, and control of phosphoinositide-specific phospholipase C. *Physiol Rev* 80: 1291-335
- Reymond N, d'Agua BB, Ridley AJ (2013) Crossing the endothelial barrier during metastasis. *Nat Rev Cancer* 13: 858-70
- Rhee SG (2001) Regulation of phosphoinositide-specific phospholipase C. *Annual review of biochemistry* 70: 281-312
- Shi L, Zhao M, Luo Q, Ma YM, Zhong JL, Yuan XH, Huang CZ (2010) Overexpression of PIP5KL1 suppresses cell proliferation and migration in human gastric cancer cells. *Mol Biol Rep* 37: 2189-98
- van Horck FP, Lavazais E, Eickholt BJ, Moolenaar WH, Divecha N (2002) Essential role of type I(alpha) phosphatidylinositol 4-phosphate 5-kinase in neurite remodeling. *Curr Biol* 12: 241-5
- Vo YP, Hutton JC, Angleson JK (2004) Recycling of the dense-core vesicle membrane protein phogrin in Min6 beta-cells. *Biochem Biophys Res Commun* 324: 1004-10
- Wuttke A, Sagetorp J, Tengholm A (2010) Distinct plasma-membrane PtdIns(4)P and PtdIns(4,5)P2 dynamics in secretagogue-stimulated beta-cells. *J Cell Sci* 123: 1492-502
- Yamazaki M, Miyazaki H, Watanabe H, Sasaki T, Maehama T, Frohman MA, Kanaho Y (2002) Phosphatidylinositol 4-phosphate 5-kinase is essential for ROCK-mediated neurite remodeling. *J Biol Chem* 277: 17226-30

2nd Editorial Decision

28 September 2015

Thank you for submitting your revised manuscript for consideration by the EMBO Journal. It has now been seen by two of the original referees whose comments are enclosed. As you will see, a final revision is needed prior to acceptance here.

Referee #1 requests a better description of the ELISA used to measure cellular PI4,5P2 levels.

Referee #2 still raises several concerns regarding the proposed mechanism.

I would thus like to ask you to address the remaining issues in a final revision. Some down-toning of the conclusions with respect to cofilin as a main target of PIP2 might address the concerns of referee #2.

Please also note our guide to authors regarding the format of your revised manuscript. We have now the 'Expanded View' format for supplementary information and we encourage publication of source data that will be linked to the figures.

Figures appropriate for the Expanded View format are those of particular value to specialist readers, but which are not essential to follow the main thread of the paper for the general reader. Previously, important data would have been difficult to find and access as Supplementary Information. EMBO Press strongly encourages authors to select a limited number (typically 5) of supplementary figures for inclusion in the article proper as Expanded View figures in order to improve their accessibility, visibility and utility. Any extra figures that are not promoted to the Expanded View should be included in a 'traditional' supplementary PDF (along with supplementary text and tables) now called the Appendix.

If you would like to provide source data for your figures (e.g. excel sheets or full blots), please provide those upon resubmission as well.

I would thus like to ask you to re-format the revised version according to our guidelines, we will then be able to swiftly proceed towards publication.

Thank you for the opportunity to consider your work for publication. I look forward to your revision.

REFEREE REPORTS

Referee #1:

The authors have addressed all of this reviewer's questions experimentally and the conclusions of the study now are greatly solidified. Considering the novelty and significance of the study in the phosphoinositide field, this reviewer suggests publishing the paper in the EMBO journal if my only concern is clarified. See below.

In the rebuttal, the authors measured cellular PI4,5P2 levels by ELISA. However, it is unclear how they did this. The data indicate that only approximately 50% of cellular PI4,5P2 is detected from the membrane fraction, which is somewhat surprising. Providing detailed method (how to extract lipids, how to extract membrane fraction and how to normalize PI4,5P2 levels, by protein quantification, cell number or other lipids?) will be helpful for readers and this reviewer.

Referee #2:

In their revised manuscript, Sengelaub et al. have addressed many of my concerns. They have performed many additional experiments and overall the manuscript has improved significantly. However, the authors still have not fully addressed one of my main concerns:

1) The authors try to explain how a sustained decrease in PIP2 can lead to sustained cofilin activity. Whilst some of their arguments seem valid, I am not completely convinced by their overall conclusions. I agree that the activity of PLC and PTPR2 are spatially and temporally restricted, and that upon activation the PIP2 reduction is not complete. Activation of the proteins would lead to a temporal (less than 3 minutes) and sudden release of non-phosphorylated and active cofilin, which is quickly phosphorylated and therefore inactivated (within 3 to 5 minutes). In contrast, in the case of a sustained PIP2 decrease (hours), there is no constant release of a large pool of cofilin that can be active before it is inactivated. Therefore, in their experiments where PLCB and PTPR2 are overexpressed, a sustained (hours and even days) PIP2 decrease can never lead to constitutively active cofilin. I would even anticipate the opposite effect: a sustained decrease of PIP2 will lead to a lower number of cofilin molecules at the plasma membrane, and therefore a lower pool of cofilin that can be activated by e.g. PLCgamma. Moreover, their knockdown data is also not explained by their conclusions. If PIP2 levels are truly higher, more cofilin should be bound to the membrane and be able to be activated upon PIP2 hydrolysis by e.g. PLCgamma. Therefore it may even be expected that it would lead to more cofilin activation. Although their PIP2 data is striking, I do not believe this is mediated through cofilin activation, but rather can be explained by one of the many other proteins that can bind to this lipid.

2) The authors believe that the addition of exogenous PIP2 affects the very early stages of metastatic colonization, since PIP2 levels remain high for just one hour (which I believe is still very long considering the fast turnover rate of PIP2). The authors reason that cells need to immediately extravasate in order to enter the lungs and form metastasis, so therefore the short-term elevation of PIP2 is enough to affect metastasis. However, I do not agree with this argument. Firstly, lodged cells can also proliferate within blood vessels and form metastases without extravasation. Secondly, after one hour the PIP2 level has returned to base line and the cells will no longer experience a potential inhibition effect of extravasation. At best, the extravasation is delayed by one hour which does not explain the large inhibition in outgrowth potential.

Minor point:

The authors have confirmed their PIP2 measurements with PIP2 ELISA. This data confirms that knockdown of PLCB and PTPRN leads to more plasma membrane PIP2 and overexpression of these proteins leads to PIP2 decreases. However, this measurement shows some inconsistencies: the siCntrl is significantly lower than the Cntrl and to the same level as the PLCb1 OE and PTPRN2 OE. Moreover, the knockdowns show no significant differences compared to Cntrl. Obviously, the siCntrl can have off-targeted effects, but it seems suspicious that the off-targeted effect has the same magnitude as overexpression of PLCB and PTPR. I am not familiar with this assay and all the accompanying caveats, but I would be more convinced if the authors would have used either TLC or mass-spec to confirm their PIP2 data.

Referee #1:

The authors have addressed all of this reviewer's questions experimentally and the conclusions of the study now are greatly solidified. Considering the novelty and significance of the study in the phosphoinositide field, this reviewer suggests publishing the paper in the EMBO journal if my only concern is clarified. See below.

In the rebuttal, the authors measured cellular PI4,5P2 levels by ELISA. However, it is unclear how they did this. The data indicate that only approximately 50% of cellular PI4,5P2 is detected from the membrane fraction, which is somewhat surprising. Providing detailed method (how to extract lipids, how to extract membrane fraction and how to normalize PI4,5P2 levels, by protein quantification, cell number or other lipids?) will be helpful for readers and this reviewer.

Author's Response:

We thank the referee for their positive comments on our revision. To isolate low-density membrane fractions, we followed a protocol from a *Journal of Cell Biology* paper (van Rheenen et al., 2007), wherein the authors isolated low-density membranes and measured PIP2 levels by a dot blot Western blot. The ELISA was performed according to the manufacturer's directions (Echelon Biosciences). We normalized samples based on both cell count during seeding and protein quantification. We have updated our Methods section with more detail on these protocols as follows:

For low-density membrane isolation, cells were lysed in ice-cold lysis buffer (50 mM Tris, pH 7.5, 300 mM NaCl, 5 mM EGTA, 20 mM DTT, 1% Triton X-100). A fraction of the lysate was measured for protein content by BCA assay (Thermo) to normalize cell number. The harvested cell lysate was mixed with 60% OptiPrep (Sigma) to yield 40% OptiPrep final. The lysate containing 40% OptiPrep was transferred to the bottom of a centrifuge tube. 1 ml 30% OptiPrep was added on top of 40% OptiPrep lysate. 300 μ l of 5% OptiPrep was added on top of 30% OptiPrep. Tubes were centrifuged for 12h at 4°C at 100,000 g. Gradients were collected, and low-density membrane (top) fractions were used in further analysis.

PI(4,5)P2 was extracted from breast cancer cells and quantified using a PI(4,5)P2 Mass ELISA (Echelon Biosciences) according to the manufacturer's instructions. To extract PI(4,5)P2, cell or membrane pellet was incubated on ice for 5 minutes with 1 mL ice cold 0.5 M TCA. Samples were centrifuged at 1,000 g for 7 minutes at 4°C. The resulting pellet was washed with 1 mL 5% TCA/1 mM EDTA twice. To extract neutral lipids, 1 mL MeOH:CHCl₃ (2:1) was added, the samples were vortexed for 10 minutes, and centrifuged at 1,000 g for 5 minutes. Neutral lipid extraction was performed twice. To extract acidic lipids, 750 μ L MeOH: CHCl₃:12 N HCl (80:40:1) was added and samples were vortexed for 25 minutes followed by centrifugation for 1,000 g for 5 minutes. 250 μ L of CHCl₃ and 450 μ L of 0.1 N HCl was added to the supernatant. The sample was vortexed for 30 seconds and centrifuged at 1,000 g for 5 minutes to separate organic and aqueous phases. The organic phase was collected and dried under argon. Dried lipids were stored at -20°C and reconstituted in assay buffer immediately prior to ELISA procedure.

We thank the referee for highlighting the need to describe this in greater detail in the manuscript.

Referee #2:

In their revised manuscript, Sengelaub et al. have addressed many of my concerns. They have performed many additional experiments and overall the manuscript has improved significantly. However, the authors still have not fully addressed one of my main concerns:

1) The authors try to explain how a sustained decrease in PIP2 can lead to sustained cofilin activity. Whilst some of their arguments seem valid, I am not completely convinced by their overall conclusions. I agree that the activity of PLC and PTPR2 are spatially and temporally restricted, and that upon activation the PIP2 reduction is not complete. Activation of the proteins would lead to a temporal (less than 3 minutes) and sudden release of non-phosphorylated and active cofilin, which is quickly phosphorylated and therefore inactivated (within 3 to 5 minutes). In contrast, in the case of a

sustained PIP2 decrease (hours), there is no constant release of a large pool of cofilin that can be active before it is inactivated. Therefore, in their experiments where PLCB and PTPR2 are overexpressed, a sustained (hours and even days) PIP2 decrease can never lead to constitutively active cofilin. I would even anticipate the opposite effect: a sustained decrease of PIP2 will lead to a lower number of cofilin molecules at the plasma membrane, and therefore a lower pool of cofilin that can be activated by e.g. PLCgamma. Moreover, their knockdown data is also not explained by their conclusions. If PIP2 levels are truly higher, more cofilin should be bound to the membrane and be able to be activated upon PIP2 hydrolysis by e.g. PLCgamma. Therefore it may even be expected that it would lead to more cofilin activation. Although their PIP2 data is striking, I do not believe this is mediated through cofilin activation, but rather can be explained by one of the many other proteins that can bind to this lipid.

Author's Response:

We thank the referee for their positive comments on our revision. We agree with the referee that the mechanism of PTPRN2 and PLCB1 in promoting breast cancer metastasis is not limited to only the actions of cofilin. We agree with the referee that many other proteins that can also bind to PI(4,5)P2 and that additional protein effectors' activities could be modulated by this lipid and contribute to the effects seen. To address the referee's concerns, we have further emphasized this point both in the results section of the manuscript (pg. 10) and in the discussion section of the manuscript (pg. 17).

In terms of cofilin activity, the mechanism is based on a shift in cofilin activity on an aggregate cell population level. Cofilin is not constitutively active, but rather on average the cofilin population is less likely to be bound to the plasma membrane (in its inactive state), due to the lower amount of PI(4,5)P2 on average within the population. Cofilin activation is not a binary event, but rather a dynamic shift in the protein lifecycle such that on average across the cell population cofilin is more active, with a faster off-rate from the plasma membrane due to the reduced quantity of membrane PI(4,5)P2. We certainly agree with the referee that further mechanistic work in this area is needed and would require spatial and temporal analysis of cofilin activity and how it is modulated by the production of this lipid. Such analyses are beyond the scope of the work and will require future investigation. We also agree with the referee that we cannot exclude that the effects we see are not due to additional downstream proteins that interact with PI(4,5)P2. We have emphasized this in the manuscript. We thank the referee for these suggestions.

2) The authors believe that the addition of exogenous PIP2 affects the very early stages of metastatic colonization, since PIP2 levels remain high for just one hour (which I believe is still very long considering the fast turnover rate of PIP2). The authors reason that cells need to immediately extravasate in order to enter the lungs and form metastasis, so therefore the short-term elevation of PIP2 is enough to affect metastasis. However, I do not agree with this argument. Firstly, lodged cells can also proliferate within blood vessels and form metastases without extravasation. Secondly, after one hour the PIP2 level has returned to base line and the cells will no longer experience a potential inhibition effect of extravasation. At best, the extravasation is delayed by one hour which does not explain the large inhibition in outgrowth potential.

Author's Response:

Upon introduction of cell in to the vasculature, there is tremendous cell death due to sheer forces and detachment. A small number of cells are able to bind to the blood vessels in the lungs extravasate to form lung metastatic nodules as we see in our histology. If a reduced number of cells are able to extravasate during early time points, then this will have a profound effect on the formation of metastases as cells that are unable to invade into the tissue will be carried away by the large hemodynamic shear forces. Our laboratory has previously published on the importance of extravasation at early time points for successful metastasis (Furlow et al., 2015). In this study, cells were treated with a drug prior to injection, which acutely inhibits a channel that promotes cellular extravasation. The drug washes out very quickly, but we see large decreases in metastatic colonization by these cells since these cells were not able to extravasate at early time points. Thus, although the effects we see with increased PI(4,5)P2 are temporary, their effect at early time points post-injection can dramatically reduce the number of cells able to enter the lung and thus form metastases.

Minor point:

The authors have confirmed their PIP2 measurements with PIP2 ELISA. This data confirms that knockdown of PLCB and PTPRN leads to more plasma membrane PIP2 and overexpression of these proteins leads to PIP2 decreases. However, this measurement shows some inconsistencies: the siCntrl is significantly lower than the Cntrl and to the same level as the PLCb1 OE and PTPRN2 OE. Moreover, the knockdowns show no significant differences compared to Cntrl. Obviously, the siCntrl can have off-targeted effects, but it seems suspicious that the off-targeted affect has the same magnitude as overexpression of PLCB and PTPR. I am not familiar with this assay and all the accompanying caveats, but I would be more convinced if the authors would have used either TLC or mass-spec to confirm their PIP2 data.

Author's Response:

We apologize for the lack of clarity in this data. The knockdown and overexpression studies were performed in LM2 cells and MDA cells, respectively, not in the same cell line. The LM2 cells were *in vivo*-selected from MDA-MB-231 cells for their increased efficiency in metastatic lung colonization. The LM2 cells endogenously overexpress PTPRN2 and PLCB1 compared to the poorly metastatic MDA-MB-231 cells. Thus it is consistent that LM2 cells (with endogenous overexpression of PLCB1 and PTPRN2) transfected with siCntrl and MDA-MB-231 cells transfected with overexpression of PTPRN2 or PLCB1 would show relatively similar levels of PI(4,5)P2. Similarly, MDA-MB-231 cells transfected with control vector express lower levels of PTPRN2 and PLCB1, and would thus be expected to show similar levels of PI(4,5)P2 as compared to LM2 cells with knockdown of PTPRN2 or PLCB1. We apologize for not making this important point clearer before, and have amended both the figure to clearly indicate that these experiments were performed in separate cell lines.

REFERENCES

Furlow PW, Zhang S, Soong TD, Halberg N, Goodarzi H, Mangrum C, Wu YG, Elemento O, Tavazoie SF (2015) Mechanosensitive pannexin-1 channels mediate microvascular metastatic cell survival. *Nat Cell Biol* 17: 943-52

van Rheenen J, Song X, van Roosmalen W, Cammer M, Chen X, Desmarais V, Yip SC, Backer JM, Eddy RJ, Condeelis JS (2007) EGF-induced PIP2 hydrolysis releases and activates cofilin locally in carcinoma cells. *J Cell Biol* 179: 1247-59

3rd Editorial Decision

19 October 2015

Thank you for sending the revised version of your manuscript. I appreciate the introduced changes and the point-by-point response, and I am happy to accept your paper for publication in The EMBO Journal.